# Zeb1 controls neuron differentiation and germinal zone exit by a mesenchymal-epithelial-like transition

Shalini Singh[1], Danielle Howell[1], Niraj Trivedi[1], Ketty Kessler[2†], Taren Ong[1], Pedro Rosmaninho[3], Alexandre ASF Raposo[3], Giles Robinson[4], Martine F Roussel[5], Diogo S Castro[3*], David J Solecki[1*]

[1]Department of Developmental Neurobiology, St. Jude Children's Research Hospital, Memphis, United States; [2]Universite Denis Diderot (Paris 7), Paris, France; [3]Department of Molecular Neurobiology, Instituto Gulbenkian de Ciência Oeiras, Oeiras, Portugal; [4]Department of Oncology, St. Jude Children's Research Hospital, Memphis, United States; [5]Department of Tumor Cell Biology, St. Jude Children's Research Hospital, Memphis, United States

**Abstract** In the developing mammalian brain, differentiating neurons mature morphologically via neuronal polarity programs. Despite discovery of polarity pathways acting concurrently with differentiation, it's unclear how neurons traverse complex polarity transitions or how neuronal progenitors delay polarization during development. We report that zinc finger and homeobox transcription factor-1 (Zeb1), a master regulator of epithelial polarity, controls neuronal differentiation by transcriptionally repressing polarity genes in neuronal progenitors. Necessity-sufficiency testing and functional target screening in cerebellar granule neuron progenitors (GNPs) reveal that Zeb1 inhibits polarization and retains progenitors in their germinal zone (GZ). Zeb1 expression is elevated in the Sonic Hedgehog (SHH) medulloblastoma subgroup originating from GNPs with persistent SHH activation. Restored polarity signaling promotes differentiation and rescues GZ exit, suggesting a model for future differentiative therapies. These results reveal unexpected parallels between neuronal differentiation and mesenchymal-to-epithelial transition and suggest that active polarity inhibition contributes to altered GZ exit in pediatric brain cancers.

*For correspondence: dscastro@igc.gulbenkian.pt (DSC); david.solecki@stjude.org (DJS)

Present address: †Laboratory of Vascular Translational Science, University College London, London, United Kingdom

**Competing interests:** The authors declare that no competing interests exist.

## Introduction

Construction of the central nervous system's circuitry requires that newborn neurons exit their germinal zone (GZ), elaborate axons and dendrites, migrate to a final position and synaptically engage other neurons. Emerging evidence suggests that classic cell polarity signaling molecules, including the Numb endocytic adaptor, the Partitioning defective (Pard) polarity complex and LKB1/SAD kinases, create the cellular asymmetry required for neuronal development and circuit assembly (*Shi et al., 2003*; *Solecki et al., 2004*; *Kishi et al., 2005*; *Cappello et al., 2006*; *Rasin et al., 2007*; *Shelly et al., 2007*; *Barnes et al., 2007*; *Bultje et al., 2009*; *Hengst et al., 2009*; *Zhou et al., 2011*; *Chen et al., 2013*; *Famulski et al., 2010*). Indeed, defective neuronal polarization is proposed to underlie the pathology of some neurodevelopmental or neurodegenerative diseases, and restored polarity has been suggested as a potential therapeutic approach for syndromes involving perturbed polarity-linked mechanisms (*May-Simera and Liu, 2013*).

Given the importance of polarity for neuronal maturation events, great efforts have been made to define mechanisms that cell-extrinsically or -intrinsically control polarity during neuronal differentiation. Most current models suggest the activation of signaling cascades (*Barnes and Polleux, 2009*;

**eLife digest** During the formation of the brain, developing neurons are faced with a logistical problem. After newborn neurons form they must change in shape and move to their final location in the brain. Despite much speculation, little is known about these processes.

Neurons mature via the activity of several pathways that control the activity, or expression, of the neuron's genes. One way of controlling such gene expression is through proteins called transcription factors. At the same time, the developing neurons go through a process called polarization, where different regions of the cell develop different characteristics. However, it was not known how the maturation and polarization processes are linked, or how the developing neurons actively regulate polarization.

By studying the developing mouse brain, Singh et al. found that a transcription factor called Zeb1 keeps neurons in a immature state, stopping them from becoming polarized. Further investigation revealed that Zeb1 does this by preventing the production of a group of proteins that helps to polarize the cells.

The most common type of malignant brain tumour in children is called a medulloblastoma. Singh et al. analyzed the genes expressed in mice that have a type of medulloblastoma that results from the constant activity of a gene called Sonic Hedgehog in developing neurons. This revealed that these tumour cells contain abnormally high levels of Zeb1, and so do not take on a polarized form. However, artificially restoring other factors that encourage the cells to polarize caused the neurons to mature normally. Further investigation is now needed to find out whether the activity of the Sonic Hedgehog gene regulates Zeb1 activity, and to discover whether inhibiting Zeb1 could prevent brain tumours from developing.

Lewis et al., 2013; Funahashi et al., 2014), transcriptional networks (de la Torre-Ubieta and Bonni, 2011), or chromatin states (Hirabayashi and Gotoh, 2010; Yamada et al., 2014) promotes or maintains cell polarity in differentiated neurons. However, it remains unclear how developing neurons undergo discrete transitions during which polarity is delayed or promoted (Cooper, 2014; Singh and Solecki, 2015). As an example, maturing cortical neurons undergo enhanced polarization via a multipolar to bipolar transition, while cerebellar granule neuron progenitors (GNPs) remain unpolarized for an extended period while their progenitor pool expands during cerebellar development.

We discovered that the transcription factor Zeb1, a critical regulator of epithelial polarity (Vandewalle et al., 2009), is highly expressed in unpolarized GNPs and that its expression diminishes as these cells become polarized cerebellar granule neurons (CGNs). Developing CGNs provide an excellent model of the mechanisms regulating neurogenesis, neuronal differentiation, polarization linked to morphological maturation, and GZ exit (Hatten and Roussel, 2011; Hatten and Roussel, 2011). They also provide a model of migration mechanisms, since they undergo two migration phases: morphologically unpolarized GNPs and newly postmitotic CGNs migrate tangentially near the cerebellar surface in the external granule layer (EGL) while polarized CGNs migrate radially away from their GZ and cross the molecular layer (ML) to reside within the internal granule layer (IGL) (Hatten, 2002; Chédotal, 2010; Legué et al., 2015). In cerebellar medulloblastoma (MB), excessive or constitutive mitogenic signaling in GNPs disrupts the intricate balance of GZ exit and radial migration via unknown motility mechanisms (Goodrich et al., 1997; Kim et al., 2003; Yang et al., 2008; Ayrault et al., 2010).

Zeb1 functions in many organ systems, including muscle, lymphocytes, and nervous system (Takagi et al., 1998; Liu et al., 2008). Proliferating progenitors express Zeb1 in GZs in the developing mouse brain (Darling et al., 2003). While loss of Zeb1 function in the developing neocortex reduces proliferation in the VZ and SVZ (Liu et al., 2008), it remains unknown how Zeb1 regulates neural progenitor populations. Studies examining Zeb1 regulation of epithelial cell polarity provide insights. Zeb1 activates stemness pathways in immature, unpolarized epithelial cells and their transformed counterparts (Spaderna et al., 2008). It also controls transitions in epithelial differentiation and polarity plasticity: high Zeb1 expression inhibits epithelial differentiation and drives

cells toward epithelial-to-mesenchymal transition (EMT), while low expression allows mesenchymal-to-epithelial transition (MET). During EMT, Zeb1 acts as a transcriptional repressor that silences adherens junction (AJ) and apical-basal polarity genes (*Aigner et al., 2007*). Thus, Zeb1 simultaneously blocks differentiation, apical-basal polarity, and junction formation of epithelial cells, locking them into the mesenchymal state.

In the developing nervous system, EMT-like events have been observed in the transition of polarized radial glia to their delaminating progeny (*Rousso et al., 2012*; *Itoh et al., 2013*). Given that neuronal progeny undergo multiple polarity transitions after delamination from a radial glial cell, it is open to question how polarity is re-acquired after delamination as nascent neurons mature (*Cooper, 2014*; *Singh and Solecki, 2015*; *Barnes et al., 2008*). Do nascent neurons that undergo an EMT-like process also then transition through an MET-like process, like epithelial cells?

We have known for more than a decade that persistent Sonic hedgehog (SHH) signaling blocks GNP GZ exit, but the mechanism has remained a mystery. Here we hypothesized that MET-like events control the onset of neuronal differentiation and GZ exit, which involve cell polarity and cell-cell adhesion transitions. By using gain- and loss-of-function approaches, we found that Zeb1 is necessary and sufficient to maintain GNPs in an undifferentiated, unpolarized, transiently amplifying state within the EGL and to control the onset of their GZ exit. Zeb1 represses transcription of polarity and cell adhesion genes, such as *Pard6a, Pard3a* and *close homolog of L1 (Chl1)*. By using a functional screen, we found that restored expression of these genes rescues GNP differentiation, neurite extension, and GZ exit. Finally, we examined the link between morphogens and Zeb1 in controlling this process. We found that SHH, a potent GNP mitogen, maintains Zeb1 expression. Moreover, Zeb1 expression persists in MB tumor cells, the transformed GNP counterpart in which SHH signaling is persistently activated. Zeb1 loss-of-function or restored Zeb1 target expression rescued the GZ exit phenotype in *Patched1 (Ptch1)*-deficient GNPs, the progenitors of SHH-subgroup MB. Our findings show that CGN differentiation bears a remarkable similarity to mesenchymal-to-epithelial transition. The balance of EMT-like vs. MET-like processes and of proliferative vs. maturation processes may be a key developmental mechanism that, when disrupted, contributes to the pathological alteration of GZ exit in neurodevelopmental disorders and pediatric cancers.

## Results

### Zeb1 is expressed in GNPs and is extinguished during CGN differentiation

To test the hypothesis that MET-like events control the onset of GNP differentiation and GZ exit, we first surveyed expression of the canonical EMT regulators Snail1, Snail2, Twist, and Zeb1 in GNPs. Quantitative RT-PCR revealed that Zeb1 is the primary EMT factor expressed in GNPs during the early postnatal (P) peak of neurogenesis, and that expression diminishes as GNPs exit the cell cycle to differentiate into CGNs: at P7 *Zeb1* mRNA was 28-fold higher than the next most abundant transcription factor, *Snail1* (*Figure 1a*). Zeb1 protein expression confirmed our RNA analysis where it is expressed primarily in the EGL at P7 and greatly reduced at P15 (*Figure 1b*). At P7, Zeb1 is co-expressed with the proliferation marker Ki67 and two markers of GNP identity Siah2, and Meis1/2, and is greatly reduced in cyclin-dependent kinase inhibitory protein p27$^{Kip1}$/Cdkn1b (referred as p27 thereafter)-positive postmitotic CGNs in the inner EGL. We noted a subpopulation of Zeb1 positive cells in deeper layers of the cerebellum at P7. These cells represent a mixture of white matter interneuron or oligodendrocyte precursors as these cells also express Pax2 (*Maricich and Herrup, 1999*) or Olig2 (*Chung et al., 2013*) (*Figure 1—figure supplement 1*). In GNPs, Zeb1 mRNA expression was inversely correlated with the expression of the apical-basal polarity genes *Pard6a* and *Prkcz* (*Figure 1c*). Not only did *Pard6a* mRNA increase as CGN differentiation proceeded, but the promoter of this gene was active in individual GNPs at the border of the GZ, prior to their entry into the inner EGL (*Figure 1d*). Taken together, these results indicate that GNPs are mesenchymal-like, as they express a high level of Zeb1 and low levels of polarity genes.

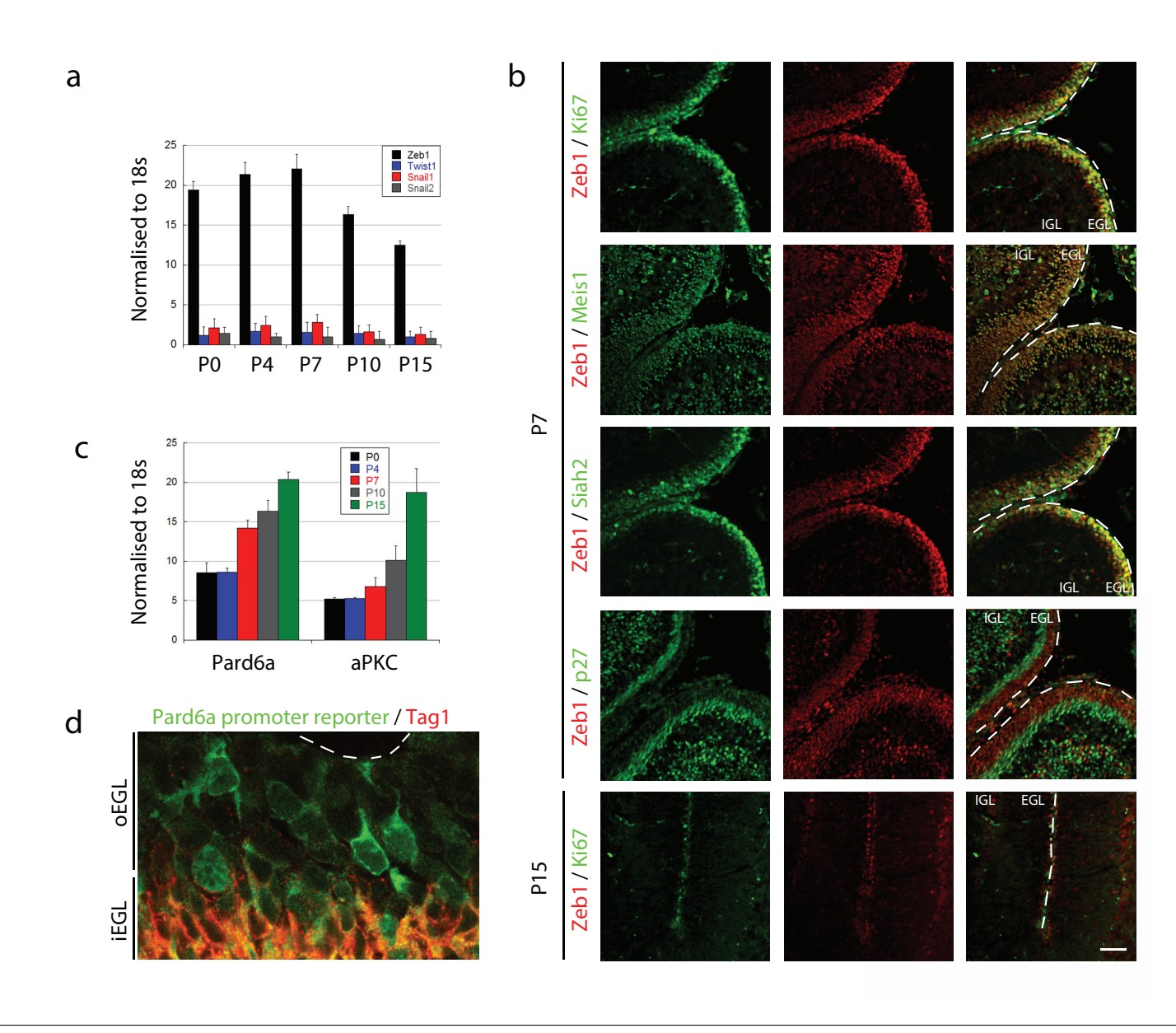

**Figure 1.** Zeb1 is the primary EMT regulator expressed in the developing cerebellum. (**a**) qRT-PCR shows that Zeb1 mRNA is more abundant than other EMT factors (Twist, Snail1, Snail2) in GNPs. Zeb1 mRNA diminishes in GNPs at P10 and P15 (Zeb mRNA was significantly different at all times, t-test p<0.01). (**b**) Immunohistochemistry in P7 and P15 cerebellum shows Zeb1 (red) GNP expression at P7 coincident with that of Ki67, Meis1/2 and Siah2 (green) but complementary to the p27Kip marker (green). Zeb1 protein diminishes at P15. (**c**) qRT-PCR shows increasing *Pard6a* and Prkcz mRNA as GNPs at P10 and P15. (**d**) Immunohistochemistry in the P7 cerebellum of *Pard6a*-EGFP BAC transgenic mice shows little *Pard6a* promoter activity (green) in the outer EGL but elevated activity in the inner EGL with TAG1-positive CGNs (red).

The following figure supplement is available for figure 1:

**Figure supplement 1.** Zeb1 is expressed in Pax2 and Olig2 positive progenitors in the developing cerebellar white matter.

## Zeb1 gain- or loss-of-function regulates CGN differentiation, neurite extension, and GZ exit

Given the Zeb1 expression profile, we reasoned that this transcription factor might regulate GNP differentiation. We used a gain-of-function approach to examine Zeb1's role in this process, as this

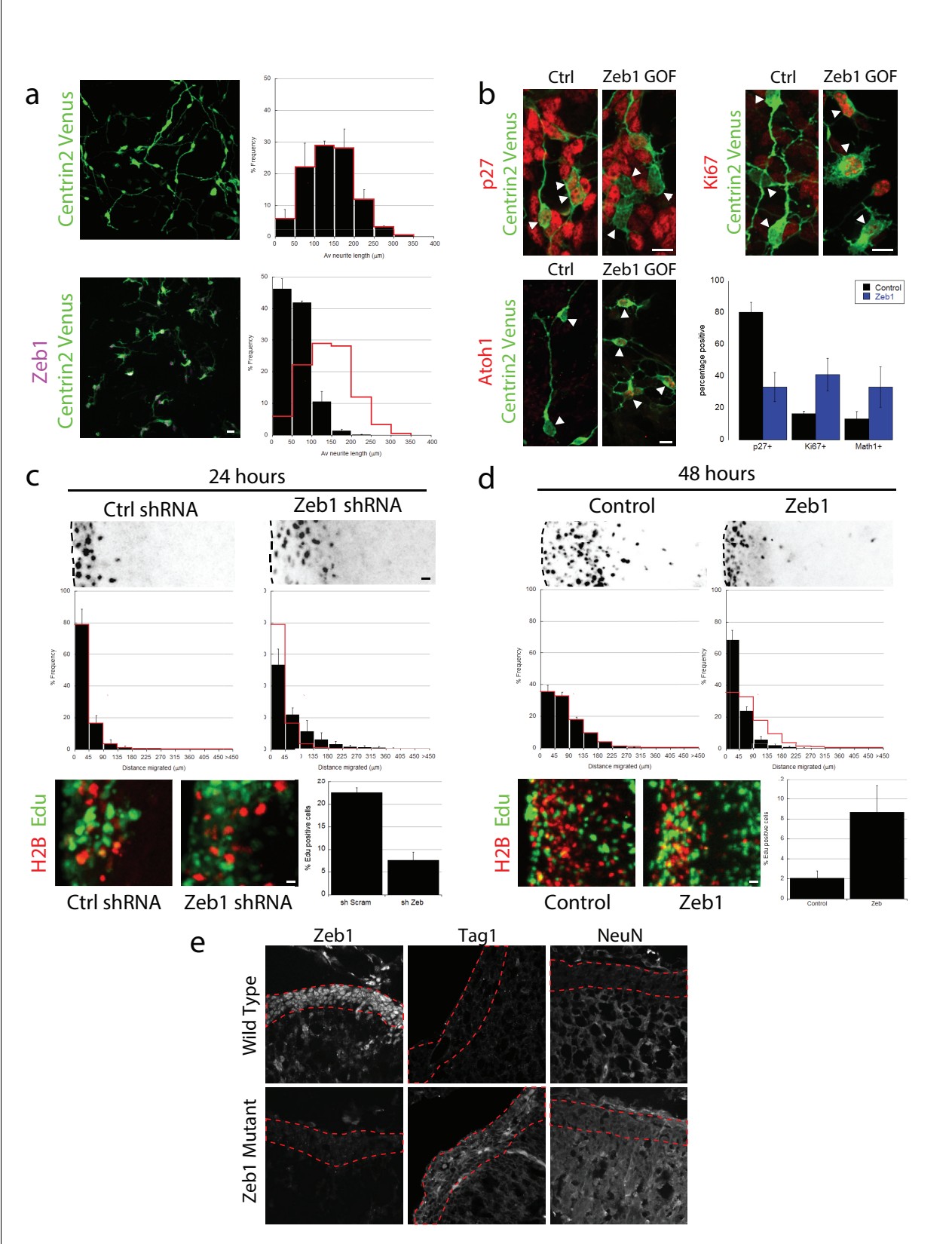

**Figure 2.** Zeb1 gain- or loss-of-function determines GNP differentiation. (**a**) Micrographs of purified CGNs nucleofected with Centrin2-Venus alone (green) or Myc-Zeb1 (magenta). After 24 hr in culture, control cells extend long neurites ($\bar{x}$ = 139.8 ± 13.3 μm. n = 1045 cells), while Zeb1-expressing

*Figure 2 continued on next page*

*Figure 2 continued*

cells have short neurites ($\bar{x}$ = 59.6 ± 3.0 μm, n = 1164 cells, $\chi^2$ test, p<0.01). (b) Micrographs of purified CGNs nucleofected with Centrin2-Venus (green) cytoplasmic marker and Myc-Zeb1. After 24 hr, levels of p27 labeling decreased, while that of Ki67 and Atoh1 increased (t-test all conditions p<0.05). C, D. P7 EGL was co-electroporated with indicated vector and H2B-mCherry. After 24 (C) or 48 (D) hr of ex vivo culture, the migration distance of labeled CGN from the pial layer (dashed line) was analyzed in 3 experiments. Histograms show migration distributions. Zeb1-silenced cells incorporated EdU at lower rates than control cells. (c) Most control shRNA-expressing cells (black) remain within the EGL (dashed lines, $\bar{x}$ = 34.2 ± 10.5 μm) at 24 hr, while Zeb1-silenced cells pre-maturely enter the ML and IGL ($\bar{x}$ = 67.5 ± 18.1 μm). (d) Control cells (black) entered the ML and IGL by 48h ($\bar{x}$ = 75.2 ± 3.5 μm), while Zeb1-expressing cells remain within the EGL ($\bar{x}$ = 40.2 ± 6.0 μm). T-tests and $\chi^2$ test showed significant differences in both conditions (p<0.01, n = 4500 to 9700 cells). (e) Immunohistochemistry in E18.5 cerebellum of wild type and Zeb1 mutant embryos shows the expected absence of Zeb1 expression in mutant embryos. Moreover, increased expression of Tag1 and NeuN differentiation markers is observed in the absence of Zeb1.

The following figure supplements are available for figure 2:

**Figure supplement 1.** In depth quantitation of slice migration assays from *Figure 2*.

**Figure supplement 2.** shRNA knockdown of Zeb1.

method maintained Zeb1 expression in GNPs and because diminished Zeb1 expression coincides with differentiation to CGNs. Purified P7 GNPs were nucleofected with an expression vector that encodes mouse Zeb1. After 1 day in vitro, control GNPs displayed features of differentiated CGNs: they extended neurites, expressed p27 and no longer expressed Ki67 and Atoh1, a marker of proliferating GNPs (*Figure 2a,b*) (*Ayrault et al., 2010*; *Flora et al., 2009*). In contrast, Zeb1-expressing cells had short, multipolar extensions ($\bar{x}$ length=140 ± 13 μm vs 60 ± 3 μm), expressed reduced p27 and sustained levels of Ki67 and Atoh1, indicating arrested maturation and proliferating, GNP-like state. While Zeb1-expressing GNPs were motile on time-lapse microscopy in dissociated cultures, they did not display the typical two-stroke nucleokinesis cycle used by differentiated CGNs and had an apolar, isotropic f-actin distribution reminiscent of GNP morphology in vivo (*Videos 1* and *2*). At the moment, it's unclear whether this mesenchymal-like morphology and random migration direction is due to a disturbed intrinsic polarity program or perturbed glial binding.

We next assessed the effect of Zeb1 function on GNP differentiation, GZ exit and migration to the IGL with the ex vivo cerebellar slice assay developed in our laboratory that specifically label GNPs (*Figure 2c and d*, see *Figure 2—figure supplement 1* for detailed analysis). We used two independent shRNA vectors to silence *Zeb1*

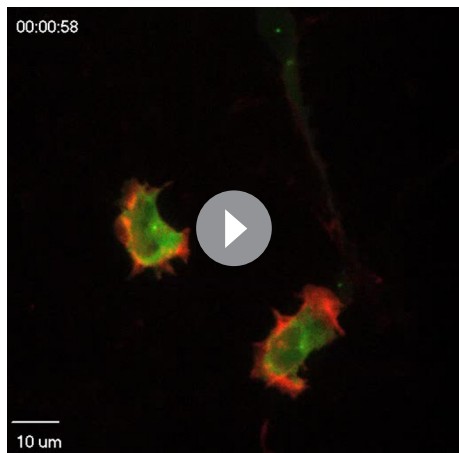

**Video 2.** Representative time lapse imaging sequence of Zeb1 over-expressing CGNs migrating in a dissociated culture labeled with Centrin2-Venus (green, centrosome) and RFP-UTRCH ABD (f-actin). The featured cells undergo random amoeboid movements with isotropic f-actin decorating the cell periphery. Note the centrosome does not adopt a polarized configuration as in *Video 1*. Time stamp= hours: minutes: seconds. Scale bar =10 μm.

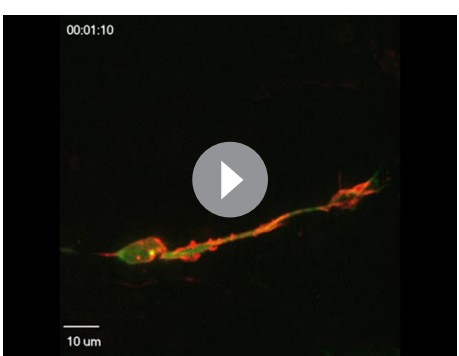

**Video 1.** Representative time lapse imaging sequence of a CGN migrating in a dissociated culture labeled with Centrin2-Venus (green, centrosome) and RFP-UTRCH ABD (f-actin). The focused cell undergoes typical two-stroke nucleokinesis with f-actin accumulation in the leading process. Time stamp= hours: minutes: seconds. Scale bar = 10 μm.

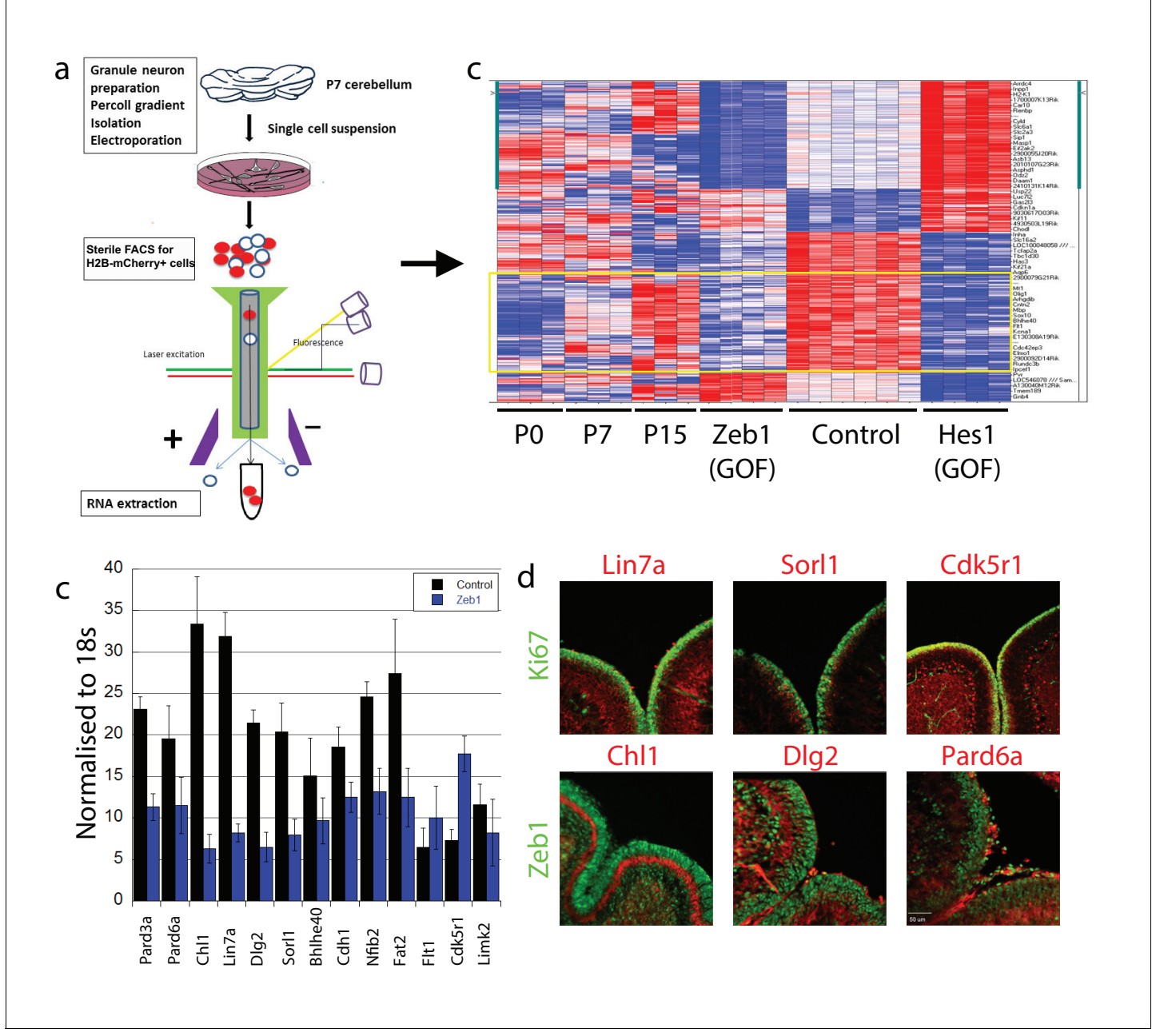

**Figure 3.** Zeb1 transcriptionally represses neuronal differentiation, cell polarity, and cell adhesion genes. (a) Schematic of procedure used to produce pure populations of CGNs for array studies. (b) Heat map of the transcriptomes of GNPs and CGNs purified from P0, P7, and P15 compared to pure populations of control (e.g. H2B-mCherry vector alone), Zeb1-expressing (e.g. H2B-mCherry and Zeb1 vector) and HES1-expressing (e.g. H2B-mCherry and HES1 vector) GNPs cultured for 24 hr in vitro. Yellow rectangle highlights genes whose expression increases with development and are repressed by Zeb1. (c) qRT-PCR shows that ectopic Zeb1 expression inhibits transcription of most of the panel of CGN differentiation markers examined. (d) Immunohistochemistry in P7 cerebellum shows Zeb1 (red) and Ki67 (green) expression complementary with expression of the Lin7a, Sorl1, Cdk5r1, Chl1, Dlg2 and Pard6a (red) CGN markers.

The following figure supplements are available for figure 3:

**Figure supplement 1.** PCA analysis of array experiments shown in *Figure 3*.

**Figure supplement 2.** PCA demonstrating purity of GNPs/CGNs prepared at different developmental stages from *Figure 3*.

**Figure supplement 3.** qRT-PCR analysis of Zeb1 target mRNA expression in GNPs or whole cerebellum.

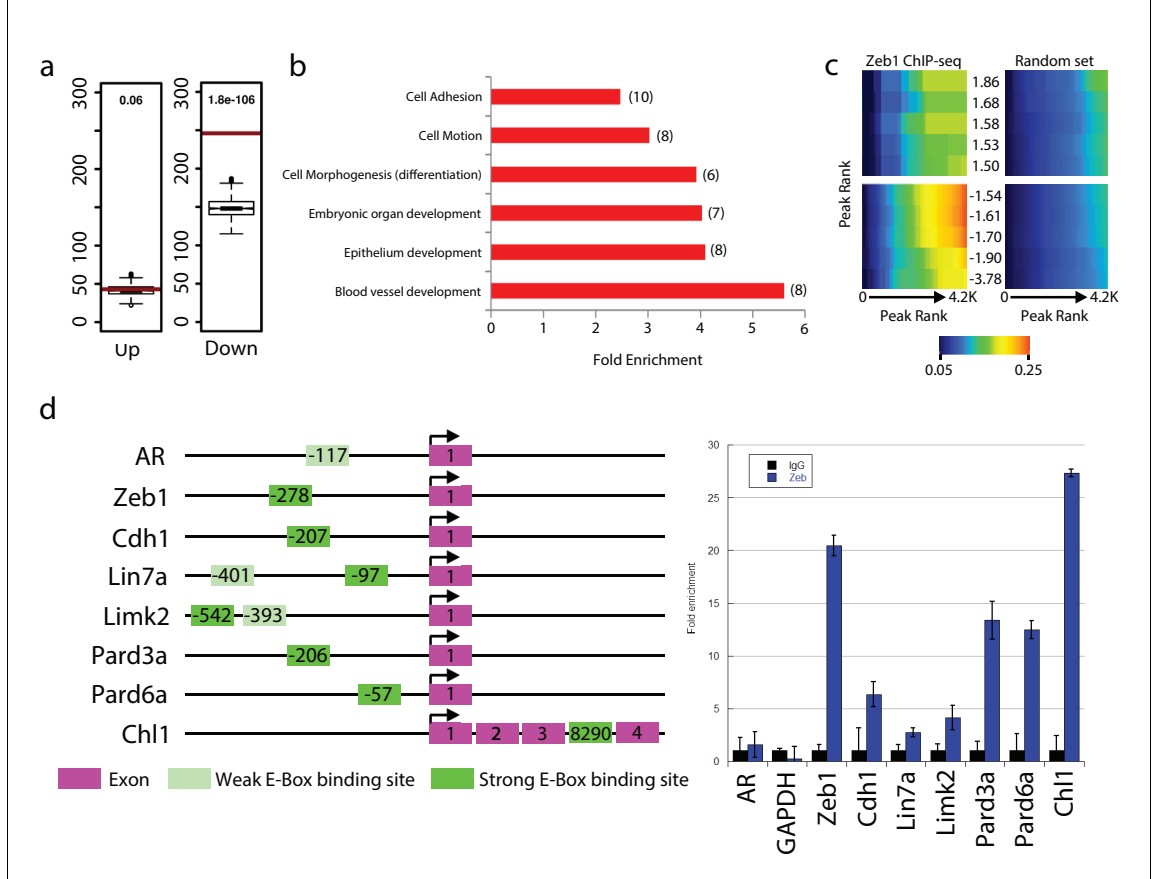

**Figure 4.** Zeb1 binds to the genomic loci of target genes identified in the expression screen. (**a**) Zeb1 binding events are significantly associated with down-regulated genes (right) but not with up-regulated genes (left) between the NS5 CHIP-Seq and CGN expression array data. Red bars: total number of binding events associated with each group of genes; boxplots: distribution of binding events associations with 1000 random sets of genes. Test data are represented as a boxplot showing the test median and 1st and 3rd quartiles; whiskers are ± 1.5 the interquartile range. (**b**) Biological processes representing clusters of gene ontology terms enriched among genes directly targeted by Zeb1. Parentheses show number of genes associated with each term. (**c**) Heat-map displaying the cumulative fraction of deregulated genes that are directly regulated by Zeb1 (up-top left panel; down-bottom left panel). Transcripts are divided in equal bins of decreasing expression fold change and plotted against Zeb1 binding events with increasing p-value. Control: 100 sets of random binding events (right panels, the mean value shown). (**d**) CHIP PCR Validation of Zeb1 binding in P7 GNPs. The schematic on the left displays gene structure. Exons are pink rectangles, Zeb1 binding unoccupied motifs are colored light green and validated Zeb1 binding sites are colored dark green. The graph on the right shows fold enrichment at the listed genes.

The following figure supplements are available for figure 4:

**Figure supplement 1.** Overview of Zeb1 ChIP-Seq dataset in NS5 neural stem cells.

**Figure supplement 2.** Annotated ChIP peaks in polarity genes and putative Zeb1 targets identified in NS5 data set.

in P7 EGL (see **Figure 2—figure supplements 1** and **2** for second shRNA migration data and validation). After 24 hr ex vivo, control EGL cells resided in the GZ and incorporated EdU, not having differentiated into CGNs or begun migrating to the IGL. In contrast, *Zeb1* silencing increased migration toward the IGL ($\bar{x}$ distance=34 ± 10 µm vs 68 ± 18 µm) and reduced EdU incorporation (22.6 ± 1.0% vs 7.6 ± 1.8% EdU positive), showing that Zeb1 loss-of-function promotes differentiation and migration toward the IGL. We next confirmed that Zeb1 activity inhibited GZ exit, using a gain-of-function approach. P7 EGL was electroporated with an expression vector for Zeb1. After 2 days ex vivo, control CGNs entered the molecular layer and IGL, while Zeb1-expressing CGNs remained within the EGL ($\bar{x}$ distance=75 ± 3 µm vs 40 ± 6 µm, **Figure 2d**) and continued to incorporate EdU (3.3 ± 0.4% vs 10.9 ± 0.1% EdU positive). To further examine the role of Zeb1 in GNP

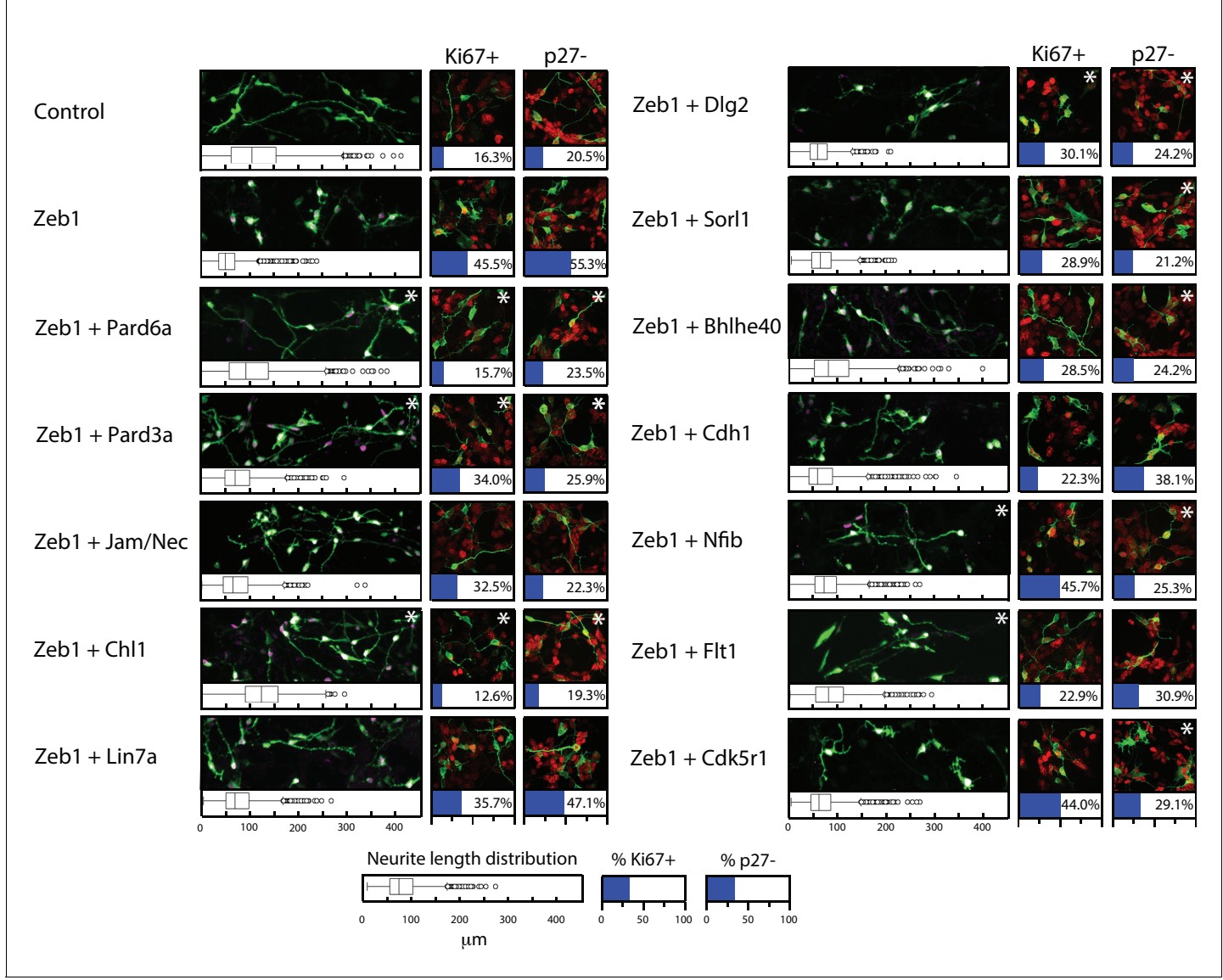

**Figure 5.** Restored expression of Zeb1-Target genes rescues neurite extension and CGN differentiation status in vitro. The rectangular images show representative morphological information and myc-Zeb1 expression; box plot below each quantifies neurite lengths in each experimental condition. On average control cells extended neurites 115.4 ± 17.7 μm [$\bar{x}$ ± sd] compared to 55.2 ± 2.6 μm. Asterisks indicate conditions significantly different to the Zeb1 data as determined by t-test (p<0.01). Images on right show representative Ki67 or p27 labeling, quantified below. Asterisks indicate statistically significant rescue of the Zeb1 phenotype by target expression determined by t-test (p<0.01).

differentiation in vivo we scored Tag1 and NeuN expression in the EGL of E18.5 Zeb1 null embryos (a time-point prior perinatal lethality observed in Zeb1 null embryos). Consistent with our ex vivo gene silencing results, loss of Zeb1 function in vivo leads to an increase in Tag1 and NeuN differentiation marker gene expression, indicating an increase of neuronal differentiation in the absence of Zeb1 (*Figure 2e*). These observations indicate that Zeb1 inhibits differentiation of GNPs to CGNs and is necessary and sufficient to restrict GNPs to their GZ niche. They also suggest that Zeb1 inhibits GNP polarization, as neurite extension, two-stroke nucleokinesis and GZ exit depend on polarity signaling complexes in CGNs.

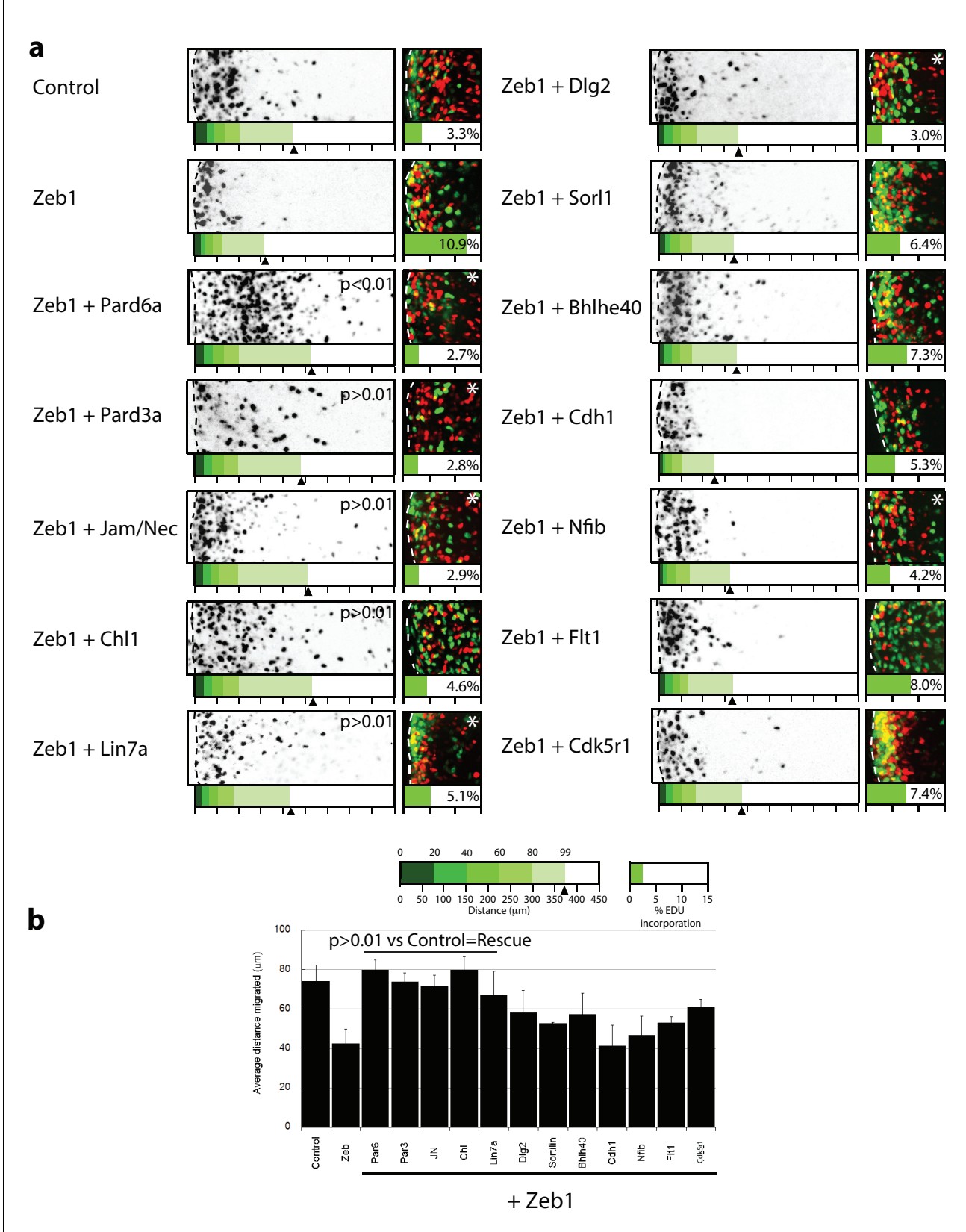

**Figure 6.** Restored expression of Zeb1 target genes rescues GNP proliferation, GZ exit and IGL-directed migration in ex vivo cerebellar slices. (a) Rectangles show representative P7 EGL slice images assessing GZ exit and IGL-directed migration. Labeled cell (black) migrate from the lateral surface
*Figure 6 continued on next page*

*Figure 6 continued*

(dashed line) to the IGL (to the right). Below each image is a cumulative distribution plot of all cells relative to a 450 µm scale. Arrowhead indicates the 99th percentile of the total population. Control cells migrated 74.0 ± 8.3 µm ($\bar{x}$ ± sd) while Zeb1 migrated 42.4 ± 7.6 µm. Images at right show representative EdU labeling with% labeling index. A statistically significant rescue of a Zeb1 phenotype in the slice migration assay is indicated by the presence of p>0.01 (t-test mean migration distance vs. control). Zeb1 and additional target expression conditions had a p-value < 0.01 vs. control indicating GZ was not rescued. Asterix indicates a statistical difference of EdU incorporation between Zeb1 and target expression condition by t-test [both p<0.01]). Reduced EdU labeling indicates a rescue of elevated proliferation in the Zeb1 gain-of-function condition. b Average migration distance shown in accompanying graph, a Student's t-test shows rescue conditions (Pard6a, Pard3a, Chl1, Jam/Nec, and Lin7a) with a p value>0.01 indicating no statistical difference from the control. Zeb1 alone and Zeb1 plus Dlg2, Sorl1, Bhlhe40, Cdh1, Nfib, Flt1 or Cdk5r1 migration differences were statistically lower than the control (t-test p<0.01), indicating GZ was not rescued with these targets.

The following figure supplements are available for figure 6:

**Figure supplement 1.** In depth quantitation of slice migration assays from *Figure 6*.

**Figure supplement 2.** Longer term ex vivo epistasis analysis.

## Zeb1 transcriptionally represses genes associated with neuronal maturation, cell polarity and cell adhesion

Having learned that Zeb1 inhibits GNP differentiation and potentially the downstream events associated with CGN polarization, we next sought to identify Zeb1 targets to determine how sustained Zeb1 expression maintains GNPs. We reasoned that as Zeb1 gain-of-function strongly inhibits GNP differentiation, it would provide a basis to identify potential Zeb1 targets. We prepared RNA from P0, P7 and P15 GNPs and used Affymetrix DNA arrays to compare their transcriptomes of these cells with those of pure, FACS-sorted GNP populations nucleofected with control, Zeb1- or HES1 expression vectors (*Figure 3a*, ArrayExpress accession number: E-MTAB-3557). We included the transcription factor HES1 because it is a known repressor of GNP differentiation downstream of the Notch2 receptor (*Solecki et al., 2001*). GNPs were selected for our developmental expression analyses as it is well established this transiently amplifying progenitor population expresses early CGN differentiation markers; such as TAG1, L1, NRCAM, NeuroD1 or TIS21 prior to their final cell cycle

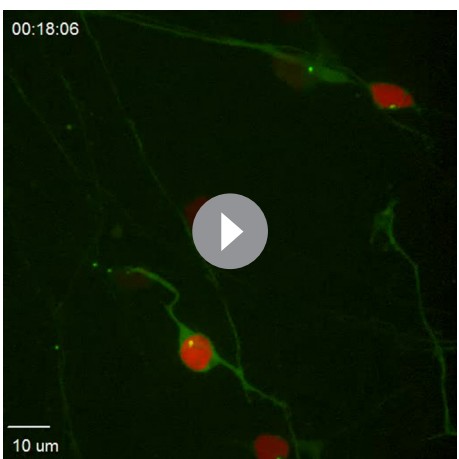

**Video 3.** Representative time lapse imaging sequence of control CGNs labeled with Centrin2-Venus (green, centrosome) and H2B-mCherry (nucleus) in a dissociated culture. The migrating cells in the field undergo typical two-stroke nucleokinesis with centrosome entering the leading process prior to somal translocation. Note: even stationary cells extend long neurites. Time stamp= hours: minutes: seconds. Scale bar= 10 µm.

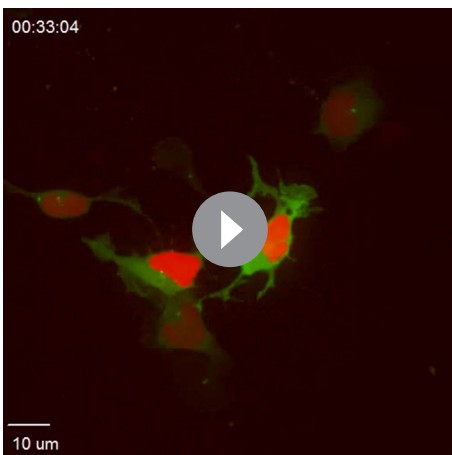

**Video 4.** Representative time lapse imaging sequence of Zeb1 over-expressing CGNs labeled with Centrin2-Venus (green, centrosome) and H2B-mCherry (nucleus) in a dissociated culture. The migrating cells in the field undergo random amoeboid movements where the centrosome adopts an unpolarized position in the cell body. Note: even stationary cells extend do not extend long neurites. Time stamp = hours: minutes: seconds. Scale bar = 10 µm.

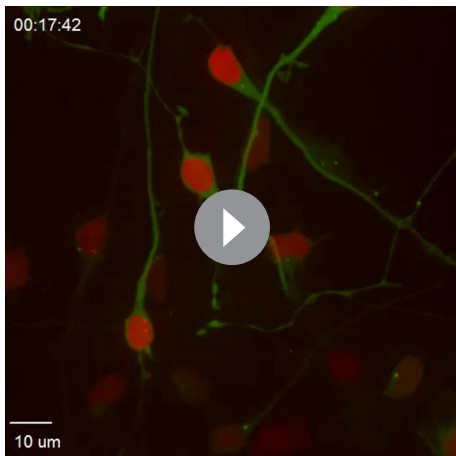 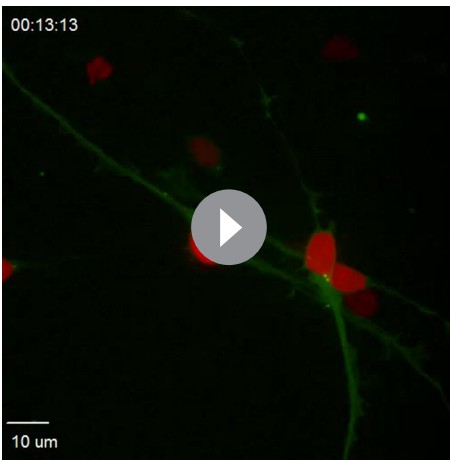

**Video 5.** Representative time lapse imaging sequence of Zeb1 over-expressing CGNs with restored Pard6a expression labeled with Centrin2-Venus (green, centrosome) and H2B-mCherry (nucleus) in a dissociated culture. The migrating cells in the field undergo typical two-stroke nucleokinesis with centrosome entering the leading process prior to somal translocation. Note: even stationary cell extend long neurites. Time stamp= hours: minutes: seconds. Scale bar = 10 μm.

**Video 6.** Representative time lapse imaging sequence of Zeb1 over-expressing CGNs with restored Pard3a expression labeled with Centrin2-Venus (green, centrosome) and H2B-mCherry (nucleus) in a dissociated culture. The migrating cells in the field undergo typical two-stroke nucleokinesis with centrosome entering the leading process prior to somal translocation. Note: even stationary cell extend long neurites. Time stamp = hours: minutes: seconds. Scale bar = 10 μm.

(*Miyata et al., 1999*; *Iacopetti et al., 1999*; *Xenaki et al., 2011*). Zeb1 gain-of-function suppressed a group of genes increasingly expressed between P0 and P15 (*Figure 3b*, *Figure 3—figure supplement 1*), consistent with previous observations that Zeb1 acts as a transcriptional repressor. Gene ontology analysis revealed this group of genes to be associated with tissue morphogenesis, epithelial polarization, cell adhesion and control of cell motility. Key members of the apical or basolateral polarity pathways (*Pard6a Pard3a, Dlg2 and Lin7a*) and *Cdh1* AJ adhesion molecule were among the Zeb1-repressed genes. In parallel, we analyzed the EMT/MET signature upon Zeb1 gain-of-function in GNPs, using a pathway-focused PCR array. Various genes previously shown to be induced during EMT were enriched in these GNPs, while a class of MET-related genes were repressed (see Tables in *Supplementary file 1B*). For further validation we selected a group of genes that included polarity complex genes (*Pard6a, Pard3a, Dlg2 and Lin7a*), cell adhesion genes (*Cdh1 and Chl1*), transcription factors associated with cell differentiation (*Bhlhe40 and Nfib*), and three randomly selected genes (*Sorl1, Flt1, and Cdk5r1*), most of which were not significantly repressed by HES1. Not only were many of these genes increasingly expressed in the normal developmental time course (*Figure 3—figure supplement 2*) and validated as suppressed in Zeb1-expressing GNPs (*Figure 3c*), the protein expression of many of them was mutually exclusive with Zeb1 or Ki67 in vivo (*Figure 3d*). A previous study in our laboratory demonstrated a similar expression profile for Pard3a (*Famulski et al., 2010*). These results suggest that many of the putative targets identified are bona fide CGN differentiation markers expressed at low levels in early postnatal GNPs. Increased polarity gene expression in differentiated CGNs, their mutually exclusive expression with GNP markers, and their suppression by Zeb1, further suggest a parallel between GNP differentiation and MET.

We next sought to investigate whether Zeb1 directly regulates genes differentially expressed in our array. As a prelude, we first assessed global Zeb1 binding sites in a ChIP-seq data set from NS5 mouse neural stem cells, which, like GNPs, express high levels of Zeb1 (*Figure 4—figure supplement 1*, ArrayExpress accession number: E-MTAB-3560). Many of the proximal promoters of key apical-basal polarity genes (*Pard6a, Pard6b, Pard3a, Pard6g*) and other Zeb1-regulated genes identified in our screen (*Chl1, Limk2*) showed clear Zeb1 binding peaks, suggesting that they are direct targets (*Figure 4—figure supplement 2*). Computational analyses comparing the genome-

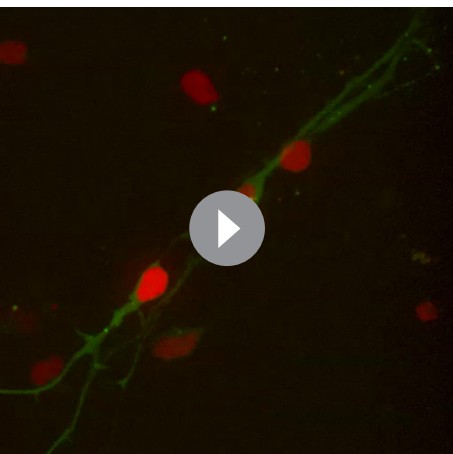

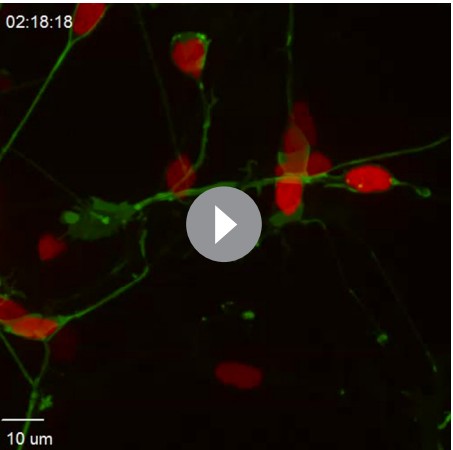

**Video 7.** Representative time lapse imaging sequence of Zeb1 over-expressing CGNs with restored Chl1 expression labeled with Centrin2-Venus (green, centrosome) and H2B-mCherry (nucleus) in a dissociated culture. The migrating cells in the field undergo typical two-stroke nucleokinesis with centrosome entering the leading process prior to somal translocation. Note: even stationary cell extend long neurites. Time stamp= hours: minutes: seconds. Scale bar = 10 μm.

**Video 8.** Representative time lapse imaging sequence of control CGNs labeled with JAM-C-pHluorin (green, adhesions) and H2B-mCherry (nucleus) in a dissociated culture. Note: exuberant cell contacts are observed among most cells. Time stamp= hours: minutes: seconds. Scale bar = 10 μm.

wide Zeb1 binding profile to the Zeb1-regulated genes identified by expression profiling showed Zeb1 binding events are highly associated with downregulated genes (*Figure 4a–c*), further pointing to Zeb1 as a transcriptional repressor in neural stem/progenitor cells. We next validated Zeb1 binding to key genes in purified P7 GNPs by ChIP PCR (*Figure 4d*). No binding was detected at non-functional regions of the genome in the *androgen receptor* and *GAPDH* genes. Weak but consistent

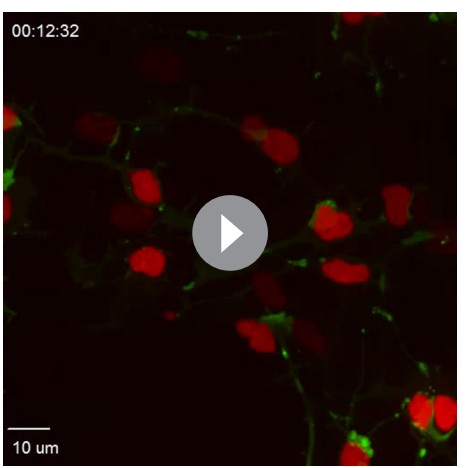

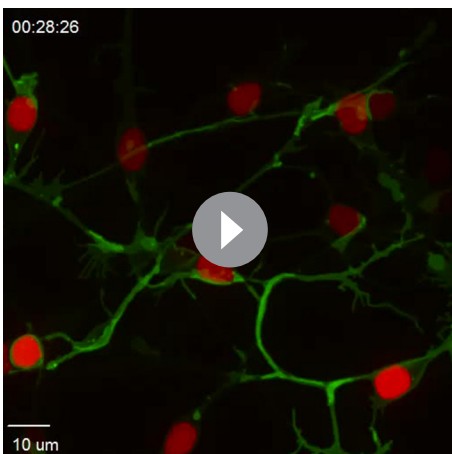

**Video 9.** Representative time lapse imaging sequence of Zeb1 over-expressing CGNs labeled with JAM-C-pHluorin (green, adhesions) and H2B-mCherry (nucleus) in a dissociated culture. Sparse cell contacts are observed among most cells. Time stamp = hours: minutes: seconds. Scale bar = 10 μm.

**Video 10.** Representative time lapse imaging sequence of Zeb1 over-expressing CGNs with restored Pard6a expression labeled with JAM-C-pHluorin (green, adhesions) and H2B-mCherry (nucleus) in a dissociated culture. Note: note cell contacts are observed among most cells. Time stamp = hours: minutes: seconds. Scale bar = 10 μm.

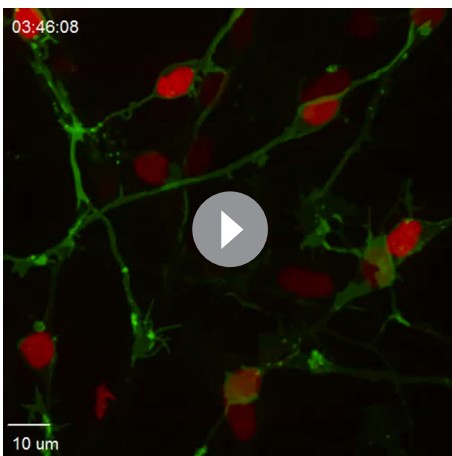

**Video 11.** Representative time lapse imaging sequence of Zeb1 over-expressing CGNs with restored Pard3a expression labeled with JAM-C-pHluorin (green, adhesions) and H2B-mCherry (nucleus) in a dissociated culture. Note: restored cell contacts are observed among most cells. Time stamp= hours: minutes: seconds. Scale bar = 10 μm.

**Video 12.** Representative time lapse imaging sequence of Zeb1 over-expressing CGNs with restored Chl1 expression labeled with JAM-C-pHluorin (green, adhesions) and H2B-mCherry (nucleus) in a dissociated culture. Note: restored cell contacts are observed among most cells. Time stamp= hours: minutes: seconds. Scale bar = 10 μm.

binding was detected in the proximal upstream regions of *Limk2* and *Lin7a*. Strong Zeb1 binding was observed at positive control regions in the *Zeb1* gene and *Cdh1* gene, at the proximal upstream sequences of *Pard6* and *Pard3a* genes and at an intronic site in the *Chl1* gene. Overall, these results indicate that Zeb1 directly regulates genes expected to play a role in cell adhesion and apical-basal polarity.

## *Par6a*, *Pard3a* and *Chl1* are *Zeb1* targets required for CGN differentiation

Given the expression profile of Zeb1 targets and the mutually exclusive expression of these genes and *Zeb1*, we postulated that some of these targets may facilitate CGN differentiation, neurite extension and GZ exit downstream of Zeb1. We individually expressed validated targets in the context of our in vitro (neurite extension, Ki67 or p27 expression status) and ex vivo (GZ exit and EdU incorporation status) Zeb1 gain-of-function assays (*Figure 2a, b and d*) to determine whether restoring individual target expression would rescue the Zeb1 phenotypes and thus functionally prioritize these targets. For this small functional screen we selected key polarity molecules (Pard6a, Pard3a, Lin7a, Dlg2), adhesion receptors (Cdh1, Chl1, constitutively active JAM-C), genes associated with cell differentiation (*Sorl1, Bhlhe40, Nfib*) and randomly selected genes (*Flt1 VEGF receptor, Cdk5r1*). Our laboratory has previously shown that Pard6a and Pard3a are required for CGN migration and GZ exit (*Solecki et al., 2004*; *Famulski et al., 2010*). Chl1 regulates neurite initiation, neuronal migration and neuronal dendrite orientation in the developing neocortex (*Demyanenko et al., 2004*; *Demyanenko et al., 2010*). Lin7 and Dlg homologs are components of the apical or basolateral polarity complexes in epithelial cells where Dlg recruits Lin7 to distinct membrane domains (*Bachmann et al., 2004*). Nfib regulates CGN differentiation (*Wang et al., 2007*), and Cdk5r1 regulates Cdk5 activity during neuronal migration (*Gupta et al., 2003*). JAM-C is not a Zeb1 target but was included because reduction of Pard3a activity reduces JAM-C adhesion and this constitutively active receptor complements CGN adhesion in the absence of Pard3a function (*Famulski et al., 2010*). Prior to the screen, we carefully titrated the quantity of expression vector needed to roughly double each target's expression in control CGNs to complement Zeb1-mediated target repression (data not shown). We observed diversity in the way individual targets modified the Zeb1 gain-of-function phenotypes in our in vitro and ex vivo assays (Supplemental Figure 5 and 6; *Figure 6—figure supplement 1*, *Supplementary file 2*). Restored expression of Pard6a, Pard3a, and Chl1 rescued all measured phenotypes to normal levels in CGNs. Individual introduction

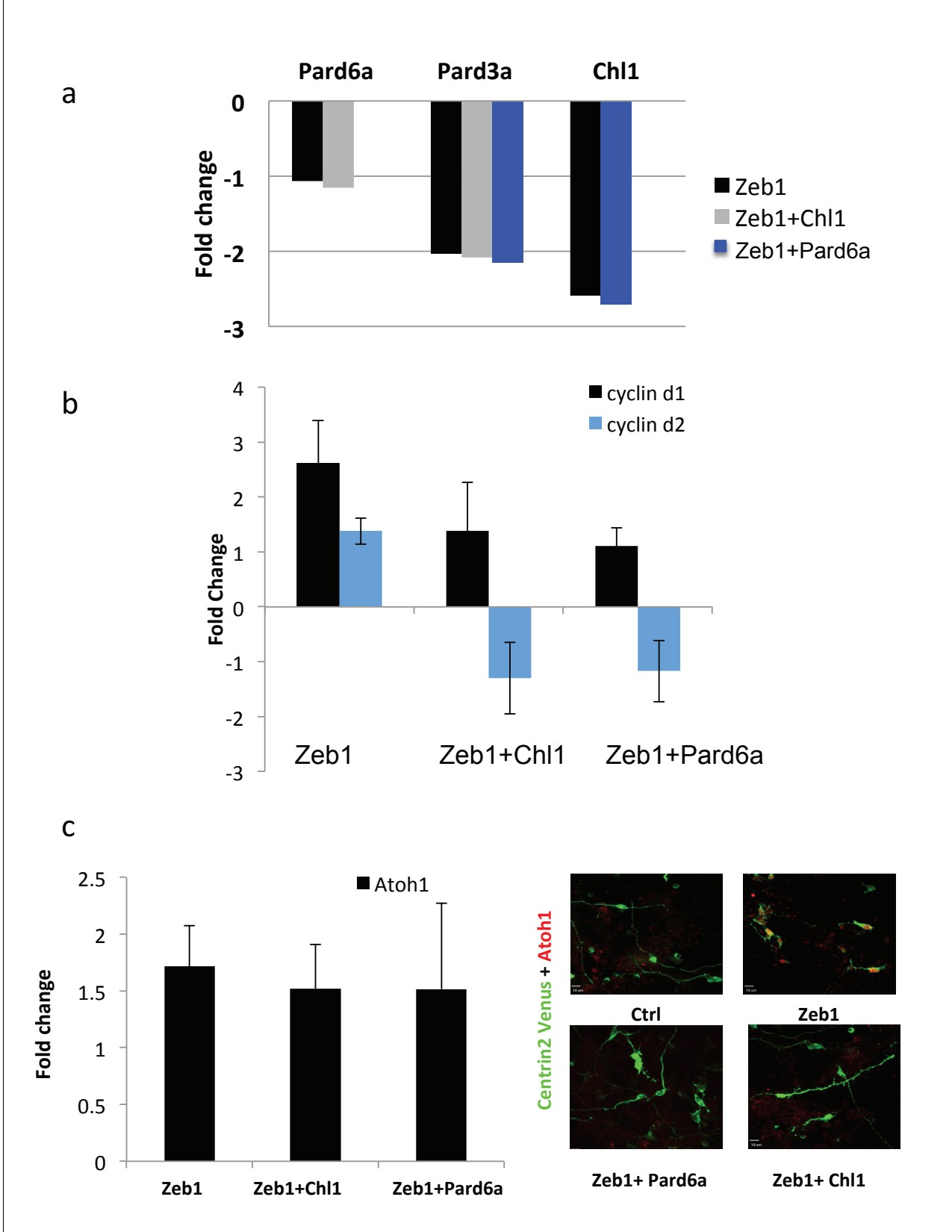

**Figure 7.** Pard6a and Chl1 rescue neuronal differentiation in the Zeb1 gain-of-function context. Cultured CGNs were nucleofected with a marker plasmid encoding H2B mCherry (or Centrin2-Venus in Panel c) alone or in combination with plasmids encoding Myc-Zeb1 plus single plasmids
*Figure 7 continued on next page*

*Figure 7 continued*

encoding *Pard6a* or *Chl1* in our array expression screen. After 24 hr in culture, nucleofected cells were FACS sorted to isolate mRNA (**a, b, c**) or stained with antibodies to highlight morphology/Atoh1 expression (**c**). a. qRT-PCR analyses shows that: 1) Pard6a and *Pard3a* expression continues to be suppressed in Chl1 rescued GNPs and 2) Pard3 and Chl1 expression continues to be suppressed by Zeb1 Pard6a rescued GNPs. NS = not shown. (**b**) qRT-PCR analyses shows that Zeb1 gain-of-function induced CyclinD1 and CyclinD2 mRNA expression and that both restored expression of Chl1 and Pard6a reduces D-type cyclin expression. (**c**) qRT-PCR analyses shows that Zeb1 gain-of-function mildly induced Atoh1 mRNA expression. While Chl1 and Pard6a rescue have little affect on Atoh1 mRNA expression, restored expression of both these genes strongly reduce Atoh1 protein expression detected by immunocytochemistry.

The following figure supplement is available for figure 7:

**Figure supplement 1.** Immunocytochemical analysis of Chl1 expression in Control, Zeb1-expressing or Pard6a and Pard3 rescued CGNs.

of each of these downstream Zeb1 targets allowed GNPs to acquire mature CGN status, characterize long neurites, expression of the p27 cell cycle inhibitor, absence of Ki67 labeling or EdU incorporation and GZ exit with subsequent migration to the IGL, even with Zeb1 gain-of-function. Constitutively active Jam-C and the basolateral polarity protein Lin7a did not influence maturation parameters in vitro (*Figure 5*); however, both stimulated cell cycle exit, GZ exit and migration ex vivo (*Figure 6*), suggesting that they act non-cell–autonomously in the complex ex vivo environment. Dlg2 stimulated cell cycle exit and p27 expression in all conditions tested but was unable to rescue neurite extension or migration ex vivo. Four genes, *Sorl1, Bhlhe40, Nfib, and Cdk5r1*, reestablished p27 expression. p27 is known for its cell cycle inhibitory and cytoskeletal regulatory properties; however, p27 expression alone in Sorl1-, Bhlhe40-, Nfib- or Cdk5r1-expressing cells was insufficient to rescue the other features of mature CGNs, such as neurite extension, loss of Ki67 labeling/EdU incorporation, or GZ exit and migration to the IGL. Restored expression of Nfib and Flt1 enhanced neurite extension but failed to rescue the full spectrum of mature CGN features, much like the genes that stimulated p27 expression. Additionally, longer term ex vivo incubations revealed that *Cdh1*, *Cdk5r1* and *Sorl1* were not sufficient to rescue IGL-directed migration of Zeb1 over-expressing cells (*Figure 6—figure supplement 2*). Interestingly, *Bhlhe40* expression, a negative regulator of EMT could rescue with a 72 hr ex vivo incubation. Time-lapse imaging revealed of cultured neurons revealed that Pard6a-, Pard3a-, and Chl1-rescue also restored two-stroke nucleokinesis and JAM-C adhesion levels, two cell biological outputs of the PAR complex function in maturing CGNs (*Solecki et al., 2004*; *Famulski et al., 2010*) (see *Videos 3–12*).

Given that *Pard6a* and *Chl1* were among the targets whose restoration most potently rescued Zeb1 gain-of-function phenotypes, we sought mechanistic insight into this rescue by further characterizing expression of key factors in proliferating GNPs, D type Cyclins and Atoh1. Restored Pard6a and Chl1 expression did not affect the levels at which Zeb1 suppressed its target genes, indicating that Pard6a and Chl1 did not counteract Zeb1 at the transcriptional level or non-specifically reduce Zeb1 target repression in our assay system (*Figure 7a*). Moreover in the case of Chl1 protein, restored Pard6a and Pard3 did not rescue Chl1 expression as assayed by immunocytochemistry in dissociated CGNs (*Figure 7—figure supplement 1*). Restored Pard6a or Chl1 expression reduced Zeb1-mediated activation of CyclinD1 and CylinD2 mRNA and Atoh1 protein levels, all of which are required to maintain GNPs in the undifferentiated state (*Figure 7b and c*) (*Ayrault et al., 2010*; *Flora et al., 2009*; *Huard et al., 1999*). The broad rescue of Zeb1 gain-of-function phenotypes by the Pard6a and Pard3a polarity proteins and the Chl1 adhesion molecule demonstrates that these Zeb1-supressed targets are prerequisites for mature CGN characteristics. These findings also reinforce the parallel between CGN differentiation and polarity regulation in cells of epithelial origin. Not only do Zeb1 and polarity proteins show mutually exclusive expression in GNPs and CGNs, but the functional screen also shows that their functional antagonism regulates the balance between the GNP and CGN states.

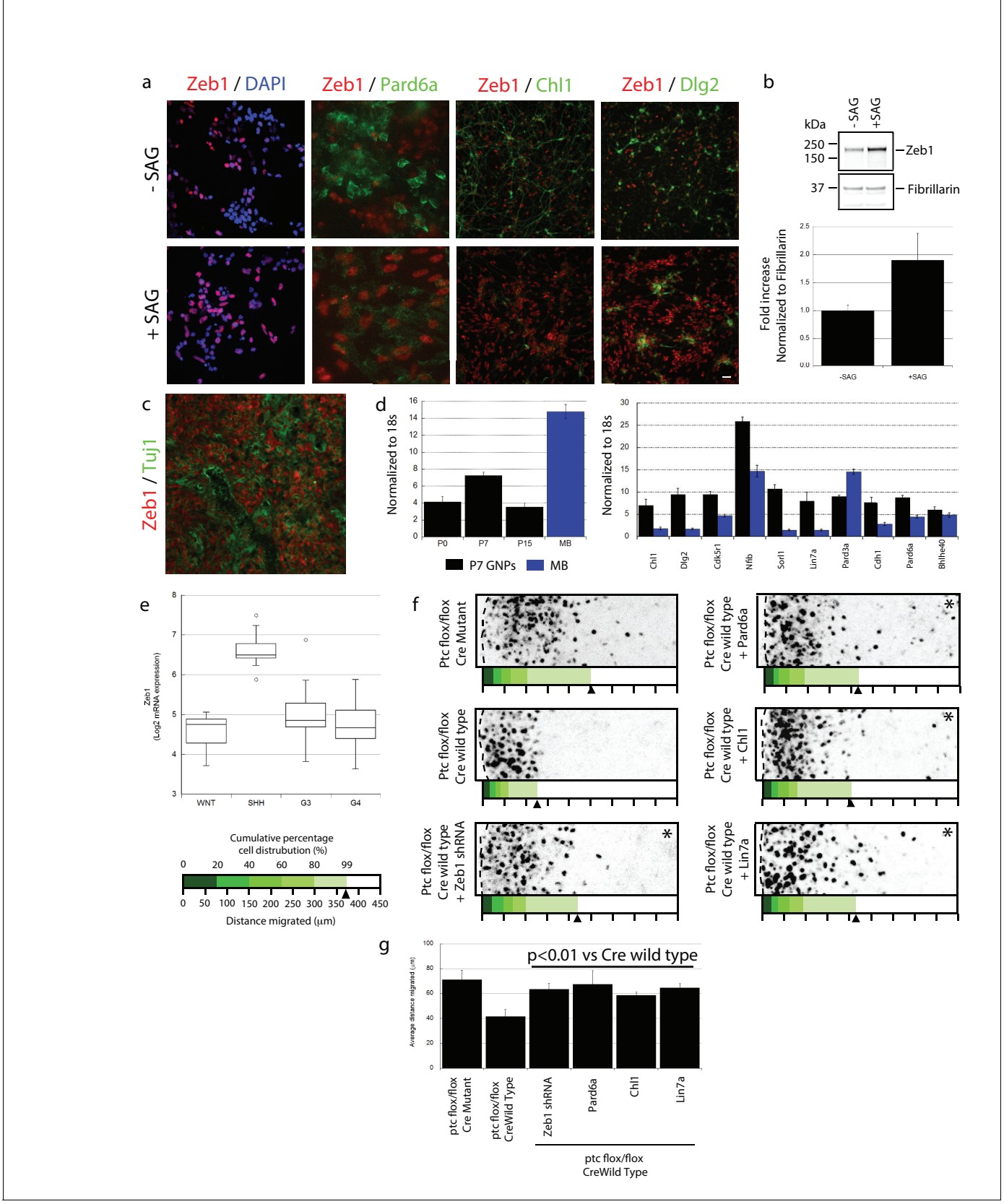

**Figure 8.** Zeb1 expression is linked to SHH signaling, and restoring polarity of Ptch1-deficient GNPs rescues GZ exit. (a) GNPs were cultured in the presence or absence of SAG, a small-molecule agonist of SHH, fixed and stained for DAPI (blue), Zeb1 (red) or the Zeb1 targets Pard6a, Chl1 and Dlg2. *Figure 8 continued on next page*

*Figure 8 continued*

Zeb1 expression was maintained, but Zeb1 target expression diminished. (**b**) Western blotting with anti-Zeb1 confirmed that Zeb1 expression was maintained in the presence of SAG. Fibrillarin was loading control (t-test, p<0.01). (**c**) Immunohistochemistry shows maintained expression of Zeb1 (red) in a Ptch1+/-, Cdkn2c-/- SHH-type mouse MB; Zeb1 expression is complementary to Tuji1 staining (green). (**d**) qRT-PCR of mRNA from Ptch1+/-, Cdkn2c-/- mouse MBs shows much higher Zeb1 mRNA expression in MB cells than in P7 GNPs. Most of the targets in our screen are expressed at a lower level in SHH MB than in P7 GNPs. (**e**) *Zeb1* mRNA expression in 4 MB subgroups. Data set includes 74 MBs (WNT n = 8; SHH n = 11; G3 n = 17; G4 n = 38) profiled on the Affymetrix U133plus2 array. (**f**) The migration distance of CGNs (black dots) from the pial layer (dashed line) was analyzed (n = 8,800 to 11,300 cells). Control cells expressing catalytically inactive Cre enter the ML and IGL (71.2 ± 7.8 μm [$\bar{x}$ ± sd]), while Ptch1-deficient GNPs expressing wild-type Cre remain within the EGL (41.4 ± 5.8 μm). Zeb1 silencing and restored expression of Pard6a, Chl1 and Lin7a rescued the defective GZ exit (Asterisks indicate conditions where rescue observed ($\chi^2$ test vs Cre mutant, p>0.8; t-test vs Cre WT, p<0.01). Below each image is a cumulative distribution plot showing the area relative to a 450 μm scale. Arrowhead indicates 99th population percentile. (**g**) Average migration distance shown in accompanying graph, a Student's t-test shows rescue conditions with a p value <0.01 vs Cre wild type.

The following figure supplement is available for figure 8:

**Figure supplement 1.** In depth quantitation of slice migration assays from *Figure 8*.

## Zeb1 is regulated by SHH, highly expressed in SHH-subgroup medulloblastoma and functionally required to retain Ptch1-deficient GNPs in the GZ

Having found that Zeb1 controls GNP differentiation and GZ exit by regulating neuronal polarity and adhesion, we next sought to identify factors that can regulate Zeb1 in GNPs. We reasoned that SHH, the required mitogen for GNP proliferation, may regulate Zeb1 expression given that it not only stimulates progenitor proliferation but also blocks CGN differentiation (*Wechsler-Reya and Scott, 1999*). GNP cultures treated with SAG, a potent small-molecule SHH agonist, displayed not only elevated Zeb1 but also decreased Pard6a and Chl1 proteins (*Figure 8a,b*). These results suggest that Zeb1 and some of its targets act downstream of the SHH signaling cascade.

The SHH pathway is activated in both mouse and human MBs derived from GNPs (*Pomeroy et al., 2002*; *Lee et al., 2003*; *Robinson et al., 2012*; *Northcott et al., 2012*). As SHH activation led to elevated Zeb1 in normal GNPs, we next examined the expression levels of Zeb1 and its targets in a mouse SHH MB model from *Ptch1 +/-, Cdkn2c -/-* mice in which SHH signaling is constitutively activated (*Uziel et al., 2005*). Unlike normal P15 cerebellum (*Figure 1b*), MBs from adult *Ptch1+/-, Cdkn2c-/-* mice displayed high levels of Zeb1 expression (*Figure 8c*). MBs contain subpopulations of cells that can proceed with neuronal differentiation. Zeb1 expression was complementary with that of class III beta-tubulin/Tuj1, an early neuronal differentiation marker, indicating that Zeb1 expression is extinguished in both normal and tumor-derived cells proceeding toward the differentiated phenotype. We also quantified the RNA expression of Zeb1 and its targets by qRT-PCR in normal GNPs and in mouse SHH MBs. Mouse MBs contained higher levels of *Zeb1* RNA than GNPs purified at P7, the time of peak Zeb1 expression (*Figure 8d*). Moreover, most Zeb1 targets identified in our Affymetrix Gene Chip array were expressed at lower levels in mouse MBs than in P7 GNPs, with the sole exception of *Pard3a* (*Figure 8d*). To broaden our analysis outside of mouse MB, we quantified *ZEB1* RNA in human MB samples (*Robinson et al., 2012*). *ZEB1* RNA was about four times higher in the human SHH MB subgroup compared to WNT, Group3 and Group4 MBs (*Figure 8e*). These results indicate that in mouse and human MB, Zeb1 expression is elevated when the SHH pathway is activated, supporting the link we observed between SHH and Zeb1 in normal GNPs. Elevated Zeb1 expression paralleled reduced expression of the targets identified in our Zeb1 gain-of-function expression profiling, validating our findings in primary GNPs.

Pre-neoplastic GNPs show a greatly delayed GZ exit, the first overt phenotype observed in mouse MB models with chronic SHH activation (*Ptch1+/-*; *Ptch1+/-, Cdkn2c-/-*; and *Ptch1Floxed mice*) (*Goodrich, 1997*; *Yang et al., 2008*; *Uziel et al., 2005*). While there is a firm link between proliferation and delayed differentiation in pre-neoplastic GNPs, it is unknown how deregulated SHH signaling delays GZ exit. Given that Zeb1 controls GNP differentiation and GZ exit and that its expression is linked with elevated SHH signaling, we postulated that Zeb1 function, and its transcriptional repression of polarity genes, may be related to the GZ exit phenotypes of GNPs with an activated SHH pathway. We developed an ex vivo model to examine the GZ exit status of GNPs

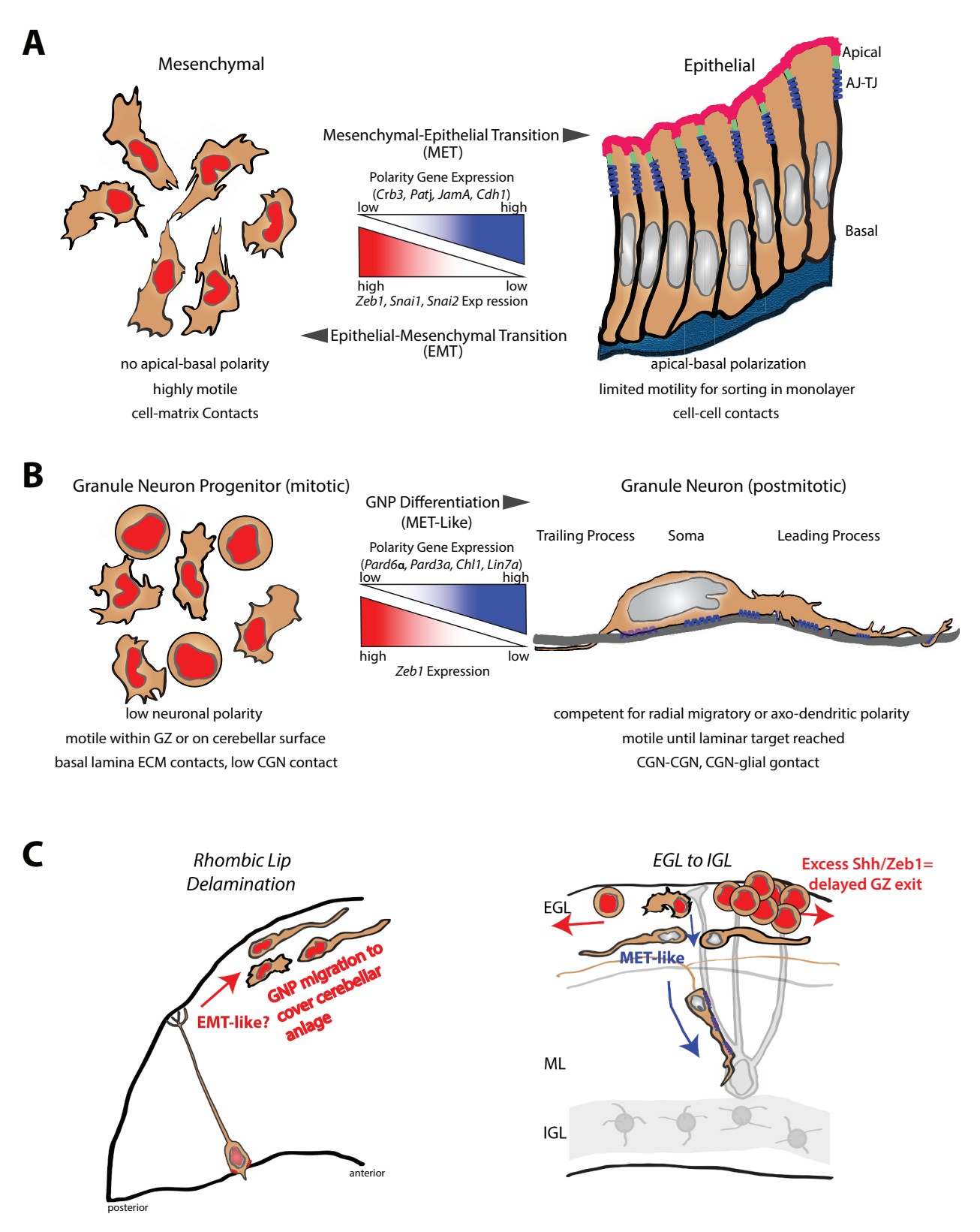

**Figure 9.** Model comparing MET to GNP differentiation. (a) Mesenchymal-epithelial transition. Left: Mesenchymal cells are nonpolar, highly motile, with prominent cell-matrix contacts. Right: epithelial cells possess apical-basal polarity. Apical membrane (pink) is separated from basolateral and basal

*Figure 9 continued on next page*

*Figure 9 continued*

membranes by tight junctions (parallel blue rectangles) and adherens (blue springs) junctions. MET-EMT balance is controlled by antagonism between transcriptional regulators and polarity genes (center panel). (b) Left: GNP. As in MET, GNPs lose Zeb1 expression as they differentiate, relieving polarity gene repression. Center panel: Change in gene expression with GNP differentiation to CGNs. Right: CGNs morphologically mature, exit their GZ and make contacts with other CGNs or glia (blue springs depict adhesion to grey glial fiber). (c) Transition from tangential migration within the EGL by GNPs and nascent CGNs to radial migration (red arrows) by polarized T-shaped CGNs is MET-like, given falling Zeb1 expression (red to grey nuclei). Blue springs depict neuron-glial adhesions. Elevated SHH signaling drives Zeb1 expression to delay GZ exit at early stage of MB tumorigenesis.

exposed to chronic SHH stimulation: we electroporated vectors encoding codon-optimized Cre recombinase or its inactive mutant into P7 cerebellar EGL from mice homozygous for *Ptch1* harboring loxP sites flanking exons 8–9 (*Ptch1* ^flox/flox^ mice) (*Ellis et al., 2003*). As the Ptch1 receptor is a negative regulator of SHH signaling, conditional *Ptch1* deletion leads to potent constitutive activation of the pathway and, over a longer time, GNP malignant transformation. GNPs expressing Cre recombinase remained largely within the EGL, but migration was unaltered by a catalytically inactive mutant ($\bar{x}$ distance = 41.4 ± 5.8 μm vs. 71.1 ± 7.8 μm; *Figure 8f*, *Figure 8—figure supplement 1*). To examine Zeb1 and Zeb1-target function in the GZ exit phenotype of *Ptch1*-deficient GNPs, we co-electroporated P7 EGL from *Ptch1* ^flox/flox^ mice with Cre recombinase and an shRNA silencing Zeb1 or vectors encoding Pard6a, Chl1 and Lin7a, which were expressed at low levels in the mouse Zeb1-expressing MB cells. *Zeb1* silencing or increased Pard6a, Chl1 and Lin7a expression restored GZ exit and migration to the IGL to near wild-type levels. Taken together, these results show that Zeb1 is functionally required downstream of SHH signaling to control GZ exit. Moreover, the MET-like transition that occurs in CGN differentiation is evident not only during normal development but also in an ex vivo model of pathological GZ exit implicated in cerebellar tumorigenesis.

## Discussion

Here we identified a key developmental mechanism of the mammalian brain wherein the onset of neuronal polarization and differentiation is restrained by Zeb1-mediated inhibition of polarity in neuronal progenitors. Conversely, diminished repression of polarity genes or adhesion receptors accompanying Zeb1 downregulation promotes morphological maturation, GZ exit, and IGL-directed migration of CGNs in the developing cerebellum. We found that Zeb1 is downregulated as GNPs begin to exit the EGL niche and that Zeb1 loss-of-function spurs precocious GZ exit and withdrawal of GNPs from the cell cycle. Failure to downregulate Zeb1 delays the onset of key polarity gene or adhesion receptor expression, morphological maturation, GZ exit, and migration to the IGL. Restoration of Pard6a, Pard3a and Chl1 expression alone is sufficient to rescue the CGN fate in the context of Zeb1 gain-of-function. These findings show that polarization is not only triggered by differentiation programs in newborn neurons but is also obstructed in transiently amplifying progenitor cells, much as polarity is regulated in epithelial-mesenchymal and mesenchymal-epithelial transitions (see *Figure 9*).

### MET and neuronal differentiation

Throughout the developing brain, newborn neurons are similarly challenged to depart their GZ niche and integrate into a functional circuit (*Hatten, 2002*; *Itoh et al., 2013*), and at each stage of their differentiation these cells must undergo reorganization of their polarity (*de la Torre-Ubieta and Bonni, 2011*; *Barnes et al., 2008*). While radial glial cells, migrating neurons and neurons elaborating axons or dendrites display a polarized morphology, transiently amplifying progenitors and newly delaminated neurons are temporarily less polarized. Conceptual parallels have been made between epithelial and neuronal polarity (*Colman, 1999*). Recently, Foxp- or Scratch-mediated inhibition of classical cadherins was shown to spur neuronal AJ loss, transition away from radial glial polarity, and delamination from the VZs of the spinal cord and cortex (*Rousso et al., 2012*; *Itoh et al., 2013*). The parallel between Foxp- and Scratch-mediated delamination of neurons and EMTs in epithelial cells is incomplete, as both delamination events may occur in postmitotic neuronal progeny. Also, we still have no clear idea how immature neurons or their progenitors transition out of their low polarity states during terminal differentiation. Our work demonstrates that transiently amplifying cerebellar

progenitors display mesenchymal characteristics, expressing high levels of Zeb1 and low levels of polarity proteins and adhesion molecules needed for maturation to CGNs.

How similar are CGN differentiation and MET? As illustrated in our model (*Figure 9*), METs are associated with acquisition of a mature, polarized morphology. Zeb1 locks GNPs into an immature morphology, just as it blocks apical-basal polarization in epithelia. Second, a common MET pattern is extensive migration followed by a final integrative positioning event (*Thiery and Sleeman, 2006*). At the population level, GNPs migrate to cover the cerebellar anlage, migrate within the EGL, and finally undergo differentiative migration to the IGL. Our results show that Zeb1 is necessary and sufficient to confine GNPs to their GZ niche, where migration is restricted to the cerebellar surface. Finally, METs involve a changing balance of cell-matrix and cell-cell contacts, in which mesenchymal cells engage in extracellular matrix adhesions and differentiated epithelial cells engage in cell-cell adhesions (*Nelson, 2009*). Similarly, early electron microscopy studies showed that GNPs remain largely contiguous with the matrix-rich pial basal lamina until they differentiate (*Hausmann and Sievers, 1985*) and develop extensive cell-cell contacts (*Rakic, 1971*; *Del Cerro and Snider, 1972*). Interestingly, we found that promotion of cell-cell contact with constitutively active JAM-C and restored Chl-1 expression rescues Zeb1 gain-of-function phenotypes. One key difference between GNP differentiation and epithelial polarity is the mir200 class of micro RNAs that inhibits Zeb1 expression in epithelial cells are not expressed in CGNs (*Uziel et al., 2009*). Overall, CGN differentiation, which is accompanied by downregulation of Zeb1, enhanced Zeb1 target expression, morphological maturation and GZ exit, bears remarkable similarity to the METs of epithelial cells as they incorporate into epithelial tissues. At the moment, it is unclear if additional EMT regulatory transcription factors behave similarly in GNP differentiation. While Zeb1 was clearly the highest expressed EMT regulatory factor relative to 18S RNA, necessity and sufficiency testing was not performed on low abundance genes like *Snai1* or *Snai2*.

We anticipate that MET associated with Zeb1 downregulation is also relevant to other brain regions. Both GNPs and cortical intermediate progenitors have delaminated from a parental radial glia, amplify transiently in a displaced GZ (EGL vs SVZ), express some similar markers (Tbr2, Id proteins, Tis21, Zeb1), and assume a simple morphologic form before differentiation. Our ChIP-seq studies show that Zeb1 occupies the promoters of polarity genes in mouse neural stem cells with telencephalic features, raising the possibility that Zeb1 may regulate the polarity of telencephalon cells. We observed that Zeb1 inhibits GNP expression of the GTPases Rnd1 and Rnd3 (data not shown), which promote VZ delamination, inhibit intermediate progenitor proliferation and enhance multipolar to bipolar transition in the neocortex, much as Zeb1 targets function in GNPs (*Heng et al., 2008*; *Pacary et al., 2011*).

## Zeb1 and neuronal polarity

Neuronal polarity regulation by Zeb1 differs from the mechanisms described in forebrain and cerebellar neurons. Neuronal polarization in the hippocampus and cortex depends on the balance of cues and signaling from extracellular, intracellular and cytoskeletal sources that shape forming axons or dendrites (*Lewis et al., 2013*). Transcriptional control mechanisms involving FOXO, SnoN1/2, NeuroD1 and NeuroD2 have been found to promote discrete stages of morphological CGN maturation, illustrating the partial dependence of axon-dendrite morphogenesis on competence that develops during differentiation (*de la Torre-Ubieta and Bonni, 2011*). Our findings show a new level of regulation of the onset of neuronal polarity in which active gene expression programs in neuronal progenitors cells can delay their competence to polarize. Thus, transiently amplifying progenitors are unpolarized not only because they do not yet express intrinsic maturation components but also because they express factors, like Zeb1, that restrain their polarization.

## Zeb1 and aberrant germinal zone exit

CGNs offer not only a model of neural development but also an excellent system to study the dysregulation of signaling pathways in disease. The best example is the link between SHH signaling, GNP proliferation, and MB tumorigenesis. Humans with activating mutations in the SHH pathway are genetically predisposed to MBs that bear many similarities to GNPs (*Raffel et al., 1997*; *Lam et al., 1999*; *Taylor et al., 2002*). Available mouse models can recapitulate SHH-associated MB (*Goodrich, 1997*; *Yang et al., 2008*; *Uziel et al., 2005*). During cerebellar development, GNPs

stream from the rhombic lip to cover the cerebellar anlage, expand clonally in the EGL in response to Purkinje cell-derived SHH, then exit mitosis and their GZ niche and migrate inward to the IGL (*Hatten et al., 1997*). When the SHH signaling pathway is deregulated in vivo, cohorts of GNPs fail to exit their GZ niche and continue to proliferate on the cerebellar surface well past the normal interval (*Goodrich, 1997*). Although migration from the mitotic niche is proposed to be linked to GNP cell cycle exit (*Choi et al., 2005*), the specific downstream GZ exit or migration mechanisms are unknown. Our finding that SHH maintains Zeb1 expression and that Zeb1 target expression is reduced in MB reveals an antagonism between the main GNP mitogen and the polarity required for GZ exit. This antagonism suggests that SHH inhibits the MET-like event we showed to control GNP GZ exit and that pre-neoplastic GNPs or MB cells are inherently polarity-deficient. The possibility that Zeb1 controls an active program to block polarization is particularly relevant to MB. These tumor cells express high levels of the FOXO and NeuroD transcription factors that promote CGN polarization, but they are insufficient to induce polarization of transformed GNPs. Thus, Zeb1 is a candidate factor that may act downstream of SHH in MB to counteract the polarization program. Finally, our results suggest future studies to determine whether restoring the polarity balance in MB will yield therapeutic benefit as a complement to existing first line- or targeted therapies.

In *Ptch1*-deficient, Zeb1-overexpressing GNPs, restored expression of selected Zeb1 targets rescues CGN differentiation, GZ exit and migration to the IGL. How do the targets, such as the PAR complex and Chl1, promote these events? In the context of Zeb1 gain-of-function, Pard6a and Chl1 expression reduced Zeb1 activation of CyclinD1, CyclinD2, and Atoh1, each of which is essential to maintain GNP proliferation (*Ayrault et al., 2010*; *Flora et al., 2009*; *Huard et al., 1999*). Thus, Pard6a and Chl1 appear to cell-intrinsically promote CGN differentiation. Consistent with this hypothesis, Pard6a and Chl1 gain-of-function in normal GNPs spurs precocious germinal zone exit (data not shown). In preliminary time-lapse imaging studies, Pard6a, Pard3a and Chl1 also rescued two-stroke motility and JAM-C adhesion levels (see *Videos 3–12*). While it is intriguing that PAR complex and Chl1 behave similarly in our functional genomics screen, further studies are necessary to clarify their potential functional interactions. Finally, an additional area of further investigation is the cooperation between transcriptional and post-transcriptional mechanisms for polarity regulation. While Pard3a is clearly transcriptionally repressed by Zeb1, it's mRNA does not display the same elevation displayed by other targets after Zeb1 expression diminishes. Interestingly, Pard3a protein expression levels is controlled by the Siah2 E3 ubiquitin ligase, thus regulation of Pard3a expression may be due to a complex interplay between transcriptional and post-translation mechanisms. In conclusion, further examination of Zeb1 function in neural progenitors and its relation to other GZ exit pathways and the MET-like conceptual model may be useful not only in understanding how normal GNPs transition to the CGN state, but also in understanding the pathogenesis of pediatric cancers linked to defective GZ exit.

## Materials and methods

### Animals

All mouse lines were maintained in standard conditions in accordance with guidelines established and approved by Institutional Animal Care and Use Committee at St. Jude Children's Research Hospital (protocol number=483). B6N.129-*Ptch1*$^{tm1Hahn}$/J strain mice were obtained from Jackson labs.

### Plasmid vectors

All cDNAs encoding protein of interest were commercially synthesized and subcloned into pCIG2 by Genscript (Piscataway, NJ, USA). Expression plasmid for Pard3a, Pard6a, Jam-C-Nectin3 and Fluorescent fusion proteins such as pCIG2 H2B-mCherry, pCIG2 RFP-UTRCH, pCIG2 Centin2-Venus and pCIG2 JAM-C-pHluorin were subcloned as previously described (*Solecki et al., 2009*).

### Preparation and nucleofection of CGNs

CGNs were prepared as described (*Hatten, 1985*). Briefly, cerebella were dissected from the brains of P7 mice and pial layer removed; the tissue was treated with trypsin/DNase and triturated into a single-cell suspension using fine-bore Pasteur pipettes. The suspension was layered onto a discontinuous Percoll gradient and separated by centrifugation. The small-cell fraction was then isolated. The

resulting cultures routinely contain 95% CGNs and 5% glia. For imaging experiments, expression vectors encoding fluorescently labeled cytoskeletal proteins and pCIG2 expressing protein of interests were introduced into granule neurons via Amaxa nucleofection, using the Amaxa mouse neuron nucleofector kit per the manufacturer's instructions and program A030. The concentration pCIG2 expression vectors used was determined such that increase in protein expression was at least two fold. After cells recovered for 10 min from the nucleofection, they were plated in either plated in 16 well slides for IHC or in movie dishes (Mattek) coated with low concentrations of poly-L-ornithine to facilitate the attachment of neurons to glial processes (according to methods established by (*Edmondson and Hatten, 1987*)

## Gene expression: RNA isolation, RT-PCR flow cytometry and Affymetrix arrays

### RNA extraction and SYBR green real-time RT-PCR

Total RNA for RT-PCR and microarray was isolated from either CGNs or whole cerebellum at different developmental time points such as postnatal day such as p0, p4, p7, p10, p15 by using the Ambion RNA Aqueous kit (Austin, TX). According to manufacturer's instructions each sample was isolated in 40 µl of elution buffer and subjected to Dnase treatment (Ambion) to get rid of any genomic contamination. Quantity and quality of the isolated RNA was checked using the Agilent 2100 Bioanalyzer with RNA 6000 Nano Chips (Agilent Technologies, Santa Clara, CA). Primer sets for each gene were designed by using Primer Express Software (Applied Biosystems, Foster City, CA) and synthesized (IDT, Coralville, IA). Sequences of the primers are listed in See Table is **Supplementary file 3A**. Two-step real-time RT-PCR was performed on the ABI PRISM 7900 Sequence Detection System by using random hexamers and the TaqMan Reverse Transcription Reagents, and the SYBR Green PCR Master Mix for the PCR step (Applied Biosystems) as described (*Singh et al., 2010*). Data were normalized by the 18S ribosomal RNA expression levels in each sample.

### Fluorescence-activated cell sorting, Affymetrix Array and analysis

To obtain a pure population of GNPs expressing the protein of interest, GNPs isolated from cerebellum of postnatal day p7 mice were nucleofected with pCIG2 H2B-mCherry (to label cells red [Control]) or mCherry with either Zeb1/Hes1, cultured for 24 hr, triturated into single cell suspension and labeled for DAPI. The viable mCherry positive cells sorting was carried out in St. Jude shared resource flow cytometry facility at St. Jude using BD Aria III SORP sorter. A bandpass 610/20 filter was used to detect mCherry signals at an excitation of 561 nm laser. The cells were directly collected in the lysis buffer and RNA was extracted as described in the previous section.

RNA for temporal developmental profiling was isolated directly after GNP isolation at time point p0, p7 and p15 as well as FACS GNPs and further analyzed by the microarray core facility at St Jude. RNA quality was determined by analysis on the Agilent 2100 Bio-analyzer, and all samples had a RIN > 8. 100 ng of total RNA was processed using the Affymetrix 3′ IVT Express Kit. Biotin-labeled cRNAs were hybridized to the Affymetrix GeneChip HT MG-430 PM array and washed, stained and scanned on the GeneTitan system (Affymetrix). Data were summarized using Affymetrix Expression Console software (v1.1) to apply the robust multi-array average (RMA) algorithm (ArrayExpress accession number: E-MTAB-3557). The arrays are RMA-normalized and batch corrected using R/ComBat. Unsupervised hierarchical clustering analysis and principal component analysis was done using Spotfire and GeneMaths. Differential expressed genes were analyzed using linear models algorithm (R/Limma). Differentially expressed genes between Zeb1/Hes1-overexpression cells and GNPs at different time point were selected using FDR corrected p-value (q value) of 0.05 and fold change of 1.5 as the cutoff. GO analysis was done using DAVID Bioinformatics Resource with the common up-regulated genes in Zeb1 and Hes1 over-expressed cells.

### RT$^2$ Profiler PCR arrays

The Mouse EMT RT$^2$ Profiler PCR Array that profiles the expression of 84 key genes was purchased from SABiosciences. Total RNA (1 µg) isolated from the flow sorted GNP isolated at p7 was used for screening by real-time PCR as per the manufacturer's instructions. Target genes whose expression

was differentially regulated (at least 2-fold difference) by Zeb1 over expression were selected and are shown in tables in *Supplementary file 1A,B*.

## Chromatin immunoprecipitation: ChIP assay

Chromatin immunoprecipitation (ChIP) was performed by using EZ ChIP reagents (Millipore) in the presence of phosphatase and protease inhibitors according to the manufacturer's instructions. Briefly, chromatin from CGNs ($\geq 1 \times 10^6$) was cross-linked for 10 min at RT with 1% formaldehyde, sonically disrupted, diluted and precleared before immunoprecipitation with either 5 µg of Zeb1 antibody or rabbit IgG as control at 4°C overnight. Protein G-agarose beads (60 µL/sample) were added and incubated for a further 1 hr at 4°C. After washing with salt gradient stringent buffers, LiCl and TE buffers, immunoprecipitated protein-DNA complexes were eluted in 200 µL of elution buffer (50 mmol/L NaHCO3, 1% SDS). Formaldehyde crosslinking was then reversed by adding 8 µL of 5 mol/L NaCl and incubating at 65°C overnight. RNA and protein were removed by sequential treatment with RNase for 30 min at 37°C and proteinase K at 45°C for 2 hr, respectively. Purified DNA fragments were then analysed with qRT-PCR using specific primer for the promoter region see Table is *Supplementary file 3B* and SYBR Green PCR Master Mix (Applied Biosystems). The results were normalised against the input control. Normalised data of three independent experiments were averaged and are presented using fold change/enrichment of each promoter region expressed as a ratio of PCR signal of samples to that of input. For example, fold increase of promoter binding is defined as the ratio of Zeb1 binding DNA compared to DNA precipitated with the IgG control antibody (set as a fixed value of 1.0).

## ChIP-seq and bioinformatics analyses

NS5 cells (*Pollard et al., 2006*) were fixed sequentially with di(N-succimidyl) glutarate and 1% formaldehyde in phosphate buffered saline (PBS) and then lysed, sonicated and immunoprecipitated with anti-Zeb1 antibody (HPA027524, Sigma), as previously described (*Castro et al., 2011*). DNA libraries were prepared from 10 ng of immunoprecipitated DNA according to the standard Illumina ChIP-seq protocol and sequenced with Illumina GAIIx. Sequenced reads were processed after mapping with SAMTools for format conversion and removal of PCR duplicates (*Li et al., 2009*) and mapped to the mouse genome (NCBI37/mm9) with Bowtie 0.12.7 (*Langmead et al., 2009*), resulting in 25 million uniquely mapped reads (ArrayExpress accession number: E-MTAB-3560). Peak calling was performed with MACS 1.4.1 *Zhang et al., 2008*) (default parameters). Profiles of genomic regions were generated using D-peaks source code (*Brohée et al., 2012*). A *de novo* search for motifs enriched at peak summits was done with Cisfinder (*Sharov and Ko, 2009*) using default parameters and a background control set of 100 bp genomic regions located 3Kp upstream input regions. Calculation of P-values for the association between binding events and deregulated genes was performed by sampling the number of genes represented in the microarray 1000 times and assuming a normal distribution. Annotation of binding events and association with genomic features was performed with PeakAnalyzer (*Salmon-Divon et al., 2010*) and the R/Bioconductor package ChIPpeakanno (*Zhu et al., 2010*).

### Cerebellar immunohistochemistry

Postnatal brains collected at p7 and p15 were fixed by immersion in 4% paraformaldehyde at 4°C for overnight followed by cryoprotection in PBS containing 30% sucrose. Histological sagittal sections were cut at 60 µm on a cryostat and pre-blocked for 1 hr in PBS with 0.1% Triton X-100 and 10% normal donkey serum. Sections were incubated overnight at 4°C with the primary antibodies followed by appropriate Alexa labeled secondary antibody (Invitrogen) at 1:1000 for an hour before mounting. Antigen retrieval was carried out for Meis1 staining.

### Immunocytochemistry of primary CGN cultures

CGNs cultured in vitro for various times were washed with PBS, permeabilized with Triton X-100 (0.1%) and blocked with normal donkey serum (10%). Primary and secondary antibody staining was carried out in PBS plus 1% normal donkey serum. The list of primary antibodies used in this study can be found in table in *Supplementary file 3C*. Alexa labeled secondary antibodies (Invitrogen)

were used to detect primary antibody stains. The slides were sealed with a coverslip using ProLong Gold mounting media (Invitrogen).

## Image and data analysis: neurite length measurement and differentiation assay

CGN cultures were imaged with a Marianas Spinning Disk Confocal Microscope (Intelligent Imaging Innovations) comprising a Zeiss Axio Observer microscope equipped with 40×/1.0 NA (oil immersion) and 63×/1.4 NA (oil immersion) PlanApochromat objectives. An Ultraview CSUX1 confocal head with 440 to 514 nm or 488/561/642 nm excitation filters and ImageEM-intensified CCD camera (Hamamatsu) were used for high-resolution imaging.

Neurite length measurements were performed using the ruler function of SlideBook software (Intelligent Imaging Innovations) by measuring the longest neurite from one end to the longest neurite on the opposite end. At least three independent biological replicates were done for each target gene. While measuring neurite length in the rescue experiments, only CGNs that showed at least two fold increase in Zeb1 expression were included for neurite measurement. Data was statistically analyzed using Microsoft Excel and graphed using Kaleidagraph v4.03.

### Ki67 and p27 data analysis

Result of Ki67 immunostaining is represented as percentage of positively stained ki67 nuclei (cut off 20–25% staining intensity) among the total number of neurons present in the image field. For Zeb1 overexpression and the epistasis studies only nuclei that showed both Zeb1 expression and ki67 were counted as a positive over total number of GNPs overexpressing Zeb1. Scoring involved counting at least 25 fields (X40 oil objective) to a minimum of 150–200 neurons in each of three independent experiments

For assessment of differentiation p27 negative cells were counted and expressed as a percentage of the total number of GNPs in the field. For analyzing p27 in the Zeb1 overexpression and epistasis experiments a scoring cut-off of 25% staining intensity for p27 and concomitant expression for Zeb1 expression was considered and counted over total number of Zeb1 overexpressing GNPs in the field.

## Ex vivo cerebellar electroporation, organotypic slice culture and imaging

P7 cerebella were dissected, soaked in endotoxin-free plasmid DNA suspended in Hanks balanced salt solution (1–5 µg/µL of each DNA was generally used, pCIG2-mCherryH2B was electroporated as a nuclear marker for migrating CGNs), transferred to a CUY520-P5 platinum block petri dish electrode (Protech International) and electroporated with a CUY21EDIT (Protech International) square wave electroporator (80 V, 5 pulses, 50 ms pulse, 500 ms interval). Electroporated cerebella were embedded in 4% low melting point agarose and 250 µm sagittal cerebellar slices were prepared using a VT1200 Vibratome (Leica Microsystems). Slices were transferred to Millicell tissue culture inserts (Millipore) and cultured in basal Eagle medium supplemented with 2 mM L-glutamine, 0.5% glucose, 50 U/ml penicillin-streptomycin, 1x B27 and 1x N2 supplements (Invitrogen) at the air-media interface for the times indicated in the Figures. In experiments that assayed proliferation, 25 µM EdU was added to culture medium and EdU incorporation was assayed by using the Click-iT assay as per manufacturer instructions (Invitrogen).

Previous characterization of this method show that greater than 97% of cell manipulated by this method are Pax6 positive CGNs in outer EGL (*Famulski et al., 2010*). For analysis of fixed specimens, slices were fixed 4% paraformaldehyde after 24 or 48 hr of culture and mounted on slides by using ProLong Gold (Invitrogen). Migration distance was measured in fixed slices by measuring the distance between the cerebellar surface and center of individual cell nuclei marked by mCherry-H2B. Central coordinates were exported from SlideBook (Intelligent Imaging Innovations) into IGOR Pro (WaveMetrics Inc.), where the distance of cells from the nearest cerebellar surface was measured and logged. Statistical analysis used Microsoft Excel and was graphed by using Kaleidagraph v4.03. For live-imaging analysis of the migration of H2B-mCherry labeled CGNs, slice cultures were transferred at 28 hr to the humidified chamber of the spinning disk confocal microscope described

above. Z-stacks (60–80 µm width, ~20 sections per stack) were collected at multiple x, y stage positions every 15 min for 24–48 hr.

## Tumor samples

qRT-PCR analyses and IHC for Medulloblastoma studies were done on GNP-like tumor cells purified from 6 different mouse tumors that developed around 20–35 week in $Ptch1^{+/-}–Ink4c^{-/-}$ mice and compared with GNPs were isolated from the cerebellum of p7 mice.

## Western blotting

P7 GNPs were cultured with or without SAG for 48 hr, and thereafter processed to obtain nuclear and cytoplasmic fractions using the Thermo NE-PER Nuclear and Cytoplasmic Extraction Reagent. Nuclear lysates were denatured using LifeTech NuPAGE sample reducing agent and LifeTech loading bufferheated to 75°C for 5 min. Samples were subjected to SDS-PAGE 4–12% Bis-Tris gel by LifeTech. The proteins were then electroblotted on to polyvinylidene fluoride membranes using an iBlot Gel Transfer Device (Invitrogen). Membrane was blocked for 1 hr at RT with Odyssey Blocking buffer diluted 1:2 and then incubated in rabbit anti Zeb1 (1:2000) (prestige) antibody and anti-Fibrillarin-loading control (1:2000) overnight at 4°C. Odyssey secondary antibodies (1:10000) was used for detecting proteins by using the Odyssey Infrared Scanner.

## Statistical analysis

All data were expressed as the mean ± SD or SE as appropriate. The Student's $t$-test was used for comparing two groups, and the one-way analysis of variance and Holm-Sidak posthoc test was used for multiple comparisons, with the level of statistical significance set at $p<0.01$ unless otherwise specified. In migration rescue assays, if rescuing conditions resulted in a $\chi^2$-test p-value $>0.8$ when compared to controls, and t-test p-value $< 0.01$ when compared to Zeb1 overexpression alone, then they were considered a rescue.

## Acknowledgements

We thank Douglas Darling for sharing Zeb1 antibodies and Chunxu Qu for analyzing Affymetrix Array data. Sharon Naron edited the manuscript. The Solecki Laboratory is funded by the American Lebanese Syrian Associated Charities (ALSAC), by grant #1-FY12-455 from the March of Dimes and by grant 1R01NS066936 from the National Institute Of Neurological Disorders (NINDS).

## Additional information

### Funding

| Funder | Grant reference number | Author |
|---|---|---|
| National Institute of Neurological Disorders and Stroke | 1R01NS066936 | David J Solecki |
| March of Dimes Foundation | #1-FY12-455 | David J Solecki |

The funders had no role in study design, data collection and interpretation, or the decision to submit the work for publication.

### Author contributions

SS, Carried out qRT-PCR, ChIP, Expression arrays, In vitro analyses and the functional screen, Conception and design, Acquisition of data, Analysis and interpretation of data, Drafted or edited the manuscript.; DH, Carried out ex vivo analyses and the functional screen, Acquisition of data, Analysis and interpretation of data, Drafted or edited the manuscript; NT, Examined Zeb1 silencing phenotypes ex vivo and prepared all Fig.s and statistical analyses, Acquisition of data, Analysis and interpretation of data, Drafted or edited the manuscript; KK, Carried out many proof of principle experiments in the initial phase of project development, Conception and design, Acquisition of data, Drafting or revising the article; TO, Performed Ptch1 fl/fl experiments and developed the Zeb1

shmir, Acquisition of data, Analysis and interpretation of data, Drafted or edited the manuscript; PR, Carried out the NS5 ChIP-seq studies and developed the Zeb1 shRNA, Acquisition of data, Analysis and interpretation of data; AASFR, Carried out bioinformatics comparison of NS5 and CGN expression data, Acquisition of data, Analysis and interpretation of data; GR, Analyzed Zeb1 in human MB, Acquisition of data, Analysis and interpretation of data; MFR, Participated in conceptual study design, provided mouse MB microarray data and coordinated mouse MB studies, Conception and design, Drafted or edited the manuscript; DSC, Designed and carried out NS5 ChIP-seq studies and designed bioinformatics comparison of NS5 and CGN expression data, Analysis and interpretation of data, Drafted or edited the manuscript; DJS, Conceived of the study, participated in its design and coordination and performed all time-lapse studies, Drafted or edited the manuscript, Acquisition of data, Analysis and interpretation of data

### Author ORCIDs
Alexandre ASF Raposo, http://orcid.org/0000-0002-2794-0508
David J Solecki, http://orcid.org/0000-0001-8481-0403

### Ethics
Animal experimentation: All mouse lines were maintained in standard conditions in accordance with guidelines established and approved by Institutional Animal Care and Use Committee at St. Jude Children's Research Hospital (protocol number = 483).

## Additional files

### Supplementary files
• Supplementary file 1. The results of mouse EMT pathway focused RT$^2$ Profiler PCR array. A fold change filtering was performed from three independent experiments using $2^{-\Delta\Delta Ct}$ (where $\Delta\Delta CT = \Delta CT$ of FACS sorted Zeb1 overexpressing GNPs–$\Delta CT$ of control GNPs) and is represented as tables. The threshold for cut off was a fold change $\geq +2.0$ or $\leq -2.0$. Supplementary Table 1A: Functional gene grouping shows an upregulation in expression of several genes such as Anhak, Col3A1, Gng11, MMP2 and3, Serpine (Pals-1) and Vim all of which are documented to be highly expressed during EMT. Supplementary Table 1B: Reciprocal expression of several key genes that are also downregulated during EMT included Dsp, Fgfbp1, Mst1r. Additionally Krt14, Nodal and Sox10 genes that are involved in differentiation showed a reduced expression when Zeb1 was overexpressed.

• Supplementary file 2. (A) Neurite length and Ki67 or p27 labeling data from *Figure 5*. (B) Average migration distance and EdU labeling data from *Figure 6*

• Supplementary file 3. (A) List of primers used for RT-PCR in microarray validation and developmental profile expression analysis. (B) Primer Sequences used for validating the ChIP -PCR studies. (C) List of Antibodies.

### Major datasets
The following datasets were generated:

| Author(s) | Year | Dataset title | Dataset URL | Database, license, and accessibility information |
| --- | --- | --- | --- | --- |
| Shalini Singh, David J Solecki | 2016 | Expression Array CGN Zeb1 and Hes1 | https://www.ebi.ac.uk/arrayexpress/experiments/E-MTAB-3557/ | Publicly available at ArrayExpress (accession no: E-MTAB-3557) |
| Pedro Rosmaninho, Alexandre ASF Raposo, Diogo S Castro | 2016 | ChIP-seq NS5 Zeb1 | https://www.ebi.ac.uk/arrayexpress/experiments/E-MTAB-3560/ | Publicly available at ArrayExpress (accession no: E-MTAB-3560) |

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
