## [Decision Letter]

Thank you for submitting your work entitled "Zeb1 controls neuron differentiation and germinal zone exit by a mesenchymal-epithelial-like transition" for consideration by *eLife*. Your article has been favorably evaluated by K VijayRaghavan (Senior editor) and three reviewers, one of whom is a member of our Board of Reviewing Editors.

The reviewers have discussed the reviews with one another and the Reviewing Editor has drafted this decision to help you prepare a revised submission.

Summary:

The paper identifies Zeb1 as a key regulator of the transition between granule cell progenitors to post-mitotic, migrating granule neurons by repressing a program of gene expression that includes cell adhesion molecules required for initiation of cell polarization and motility in the developing cerebellum.

The experiments are designed properly and the resulting data are generally convincing and well quantified. Conceptually, the idea that the transition from progenitor to post-mitotic/migratory neuron is co-opting the molecular machinery controlling mesenchymal to epithelial (and opposing the epithelial to mesenchymal transition) is appealing and rather novel. The implications of the present study for developmental neurobiology and for the molecular mechanisms underlying neuroblastoma induction could be far reaching.

Essential revisions:

1) The Zeb1-repressed genes identified are involved in cell cycle exit, migration or leading process outgrowth. It is therefore remarkable that the ectopic expression of several individual Zeb1-target genes is able to substantially rescue both cell cycle exit and migration of Zeb1-over-expressing granule neuron progenitors. (Why should an increase in Par6 make up for a deficiency in Dlg2 or surface proteins Cdh2 or Chl1?) This raises the concern that the dissociated cell and organotypic assays are not providing an accurate picture of events. The authors should perform longer term experiments (ex vivo or in vivo electroporation?) with more prolonged survivals and test which manipulations/effectors of Zeb1 are truly required for complete migration/differentiation of CGN for example. One would predict that some of the effectors are not sufficient to fully over-come the effects of Zeb1 over-expression.

2) The movies show cells that are moving in all directions. Are the cells following glia processes? Are the movies done in the ex vivo system or dissociated cultures? I am concerned whether the ex vivo migration assay is really measuring directed migration or some other process.

3) Given the authors' previous work on Siah, it is possible that the ectopic expression of one Zeb1 target might rescue expression of another Zeb1 target post-transcriptionally. As far as possible, the protein levels of Zeb1 targets should be measured in the rescue experiments (e.g., by immunofluorescence). This would show whether, for example, ectopic Par6 is rescuing Chl1 protein levels.

4) While Zeb1 appears to be the most abundantly expressed EMT gene at early postnatal stages, it isn't possible to compare different the expression of different genes using RT-PCR owing to different efficiencies of primers etc. Snail1 resembles Zeb1 in decreasing during the second postnatal week. Given the complex interactions between EMT genes, it is possible that Snail1 would similarly score as 'necessary and sufficient' in the kinds of assays done here. This should at least be discussed.

5) FACS-sorted GNPs/CGNs prepared from P0, P7, and P15 cerebellum were used as sources of RNA to identify genes that change during differentiation. However, the purity of the FACS-sorted cells was not reported. The composition of the sorted fractions should be assessed by use of markers for GNPs and CGNs. The reviewers also recommend that CGN and GNP-specific genes be identified by FACS of GFP+/- cells from pard6-EGFP GENSAT BAC transgenic mice. It would be informative to see whether expression levels of the Zeb1 target genes isolated in this study indeed increase during GNP differentiation into CGNs. Also, in other figures, it is often unclear whether RNA was isolated from purified GNPs or whole cerebellum, which would greatly affect interpretation of the data.

6) Does Zeb1 knockdown just displace transfected cells or does it also promote neuronal differentiation? The expression levels of markers for GNPs (such as Math1) and for CGNs (such as NeuroD and NeuN) should be shown.

---

## [Author Response]

*1) The Zeb1-repressed genes identified are involved in cell cycle exit, migration or leading process outgrowth. It is therefore remarkable that the ectopic expression of several individual Zeb1-target genes is able to substantially rescue both cell cycle exit and migration of Zeb1-over-expressing granule neuron progenitors. (Why should an increase in Par6 make up for a deficiency in Dlg2 or surface proteins Cdh2 or Chl1?) This raises the concern that the dissociated cell and organotypic assays are not providing an accurate picture of events. The authors should perform longer term experiments (ex vivo or in vivo electroporation?) with more prolonged survivals and test which manipulations/effectors of Zeb1 are truly required for complete migration/differentiation of CGN for example. One would predict that some of the effectors are not sufficient to fully over-come the effects of Zeb1 over-expression.*

To address this concern, we extended ourex vivoZeb1 gain-of-functionrescues assays to 72 hours post electroporation using a panel of the Zeb1 targets that did not rescue in the 48 hour experiment displayed in Figure 6 (e.g. *Bhlhe40*, *Cdh1*, *Cdk5r1* and *Sorl1*). Zeb1 gain-of-function at 72 hours still retainselectroporated cells near the cerebellar slice surface. These data are included in Figure 6—figure supplement 2. Moreover, *Bhlhe40* expression significantly restores IGL-directed migration of the context of Zeb1 gain-of-function. Interestingly, *Bhlhe40* has recently been shown to counteract epithelial-mesenchymal transitions in cancer models, suggesting that this transcription factor may possess some activity to counter Zeb1 regulation of CGN differentiation.

2) The movies show cells that are moving in all directions. Are the cells following glia processes? Are the movies done in the ex vivo system or dissociated cultures? I am concerned whether the ex vivo migration assay is really measuring directed migration or some other process.

The movies were acquired from standard granule celldissociated cultures that can be used to model radial migration of differentiated CGNs. Our laboratory and others have noted random CGN motility either when immature cells are assayed or when CGNs are arrested in an early stage of differentiation (e.g. Yamasaki et al. 2001 Development 128: 3133-44, Yacubova and Komuro 2002 J. Neuroscience 22(14):5966-81, Tarnok et al. 2005 J. Neurobiology65(2): 135-45, Famulski et al. 2010Science330(6012):1834-8). Asthe reviewers note, other potential explanations could account for the random migration phenotype. To satisfy reviewer concerns and clarify this issue in the text we now note in the manuscript there is at least two potential explanations for randomized motility 1) a disturbed intrinsic polarity program or 2) perturbed glial binding.

3) Given the authors' previous work on Siah, it is possible that the ectopic expression of one Zeb1 target might rescue expression of another Zeb1 target post-transcriptionally. As far as possible, the protein levels of Zeb1 targets should be measured in the rescue experiments (e.g., by immunofluorescence). This would show whether, for example, ectopic Par6 is rescuing Chl1 protein levels.

The reviewers raise a valid point. Given that Pard3, Pard6 and Chl1 rescued equally in all assays test, we assessed Chl1 expression in control, Zeb1 overexpressing and in the Pard6- or Pard3-rescue condition as the reviewers suggested. While we observed reduced Chl1 immunoreactivity in Zeb1 over-expressing CGNs compared to controls, Pard6 and Pard3 rescue did not enhance Chl1 immunoreactivity. These data are displayed in Figure 7—figure supplement 1.

4) While Zeb1 appears to be the most abundantly expressed EMT gene at early postnatal stages, it isn't possible to compare different the expression of different genes using RT-PCR owing to different efficiencies of primers etc. Snail1 resembles Zeb1 in decreasing during the second postnatal week. Given the complex interactions between EMT genes, it is possible that Snail1 would similarly score as 'necessary and sufficient' in the kinds of assays done here. This should at least be discussed.

We have added text to the Discussion section to discuss this possibility.

5) FACS-sorted GNPs/CGNs prepared from P0, P7, and P15 cerebellum were used as sources of RNA to identify genes that change during differentiation. However, the purity of the FACS-sorted cells was not reported. The composition of the sorted fractions should be assessed by use of markers for GNPs and CGNs.

To address specific reviewer concerns regarding the purity of our P0, P7, and P15 GNPs/CGNs, we now include additional principle component analysis (PCA) plots to serve as quality control of the entire transcriptomes of the cell populations used in these studies. This additional PCA analysis clearly show at the level of all analyzed microarray probe sets a high level of reproducibility in our P0, P7 and P15 preparations as each individual replicate of the samples used for our studies clusters closely with appropriate postnatal day matched samples. Moreover, GNP/CGN samples of all developmental stages do not cluster with purified glial cell transcriptomes indicating low levels of the most prevalent contaminating cell type in our preparations. This PCA plot is included in Figure 3—figure supplement 2 and we hope these data sufficiently address reviewer concerns regarding preparation purity.

The reviewers also recommend that CGN and GNP-specific genes be identified by FACS of GFP+/- cells from pard6-EGFP GENSAT BAC transgenic mice. It would be informative to see whether expression levels of the Zeb1 target genes isolated in this study indeed increase during GNP differentiation into CGNs.

Unfortunately we are not able to meet this request as we nolonger have access to live Pard6-EGFP GENSAT BAC transgenic mice in our laboratory. We hope the strength of our other revisions is sufficient to make up for a lack of additional data from this mouse line.

Also, in other figures, it is often unclear whether RNA was isolated from purified GNPs or whole cerebellum, which would greatly affect interpretation of the data.

We hopefully clarified this in the text sufficiently tosatisfy reviewer concerns. The only whole cerebellar RNA qRT-PCR data in the paper is restricted to Figure 3—figure supplement 3.

6) Does Zeb1 knockdown just displace transfected cells or does it also promote neuronal differentiation? The expression levels of markers for GNPs (such as Math1) and for CGNs (such as NeuroD and NeuN) should be shown.

The reviewers raise an important distinction regarding the interpretation of our ex vivoZeb1 loss-of-function experiments. Our EdU incorporation results suggestthat Zeb1-silencing promotes the postmitotic state and germinal zone exit. To address neuronal differentiation status in greater detail we now provide in vivo data from the cerebella of Zeb1 knockout embryos with NeuN and Tag1 immunostaining, two markers for CGN differentiation (postnatal analysis is impossible to perinatal lethality). Enhanced NeuN and Tag1 expression occurs in the EGL of E18.5 Zeb1 null embryos, suggesting that Zeb1 loss promotes neuronal differentiation. These data are included in a revised Figure 2.